# ETS family transcriptional regulators drive chromatin dynamics and malignancy in squamous cell carcinomas

Hanseul Yang[1†], Daniel Schramek[1†‡§], Rene C Adam[1], Brice E Keyes[1], Ping Wang[2], Deyou Zheng[2,3], Elaine Fuchs[1*]

[1]Howard Hughes Medical Institute, Robin Chemers Neustein Laboratory of Mammalian Cell Biology and Development, The Rockefeller University, New York, United States; [2]Department of Neurology, Albert Einstein College of Medicine, New York, United States; [3]Departments of Genetics and Neuroscience, Albert Einstein College of Medicine, New York, United States

*For correspondence: fuchslb@ rockefeller.edu

[†]These authors contributed equally to this work

Present address: [‡]The Lunenfeld-Tanenbaum Research Institute, Mount Sinai Hospital, 600 University Avenue, Toronto, Canada; [§]Department of Molecular Genetics, University of Toronto, Ontario, United States

**Abstract** Tumor-initiating stem cells (SCs) exhibit distinct patterns of transcription factors and gene expression compared to healthy counterparts. Here, we show that dramatic shifts in large open-chromatin domain (super-enhancer) landscapes underlie these differences and reflect tumor microenvironment. By in vivo super-enhancer and transcriptional profiling, we uncover a dynamic cancer-specific epigenetic network selectively enriched for binding motifs of a transcription factor cohort expressed in squamous cell carcinoma SCs (SCC-SCs). Many of their genes, including *Ets2* and *Elk3,* are themselves regulated by SCC-SC super-enhancers suggesting a cooperative feed-forward loop. Malignant progression requires these genes, whose knockdown severely impairs tumor growth and prohibits progression from benign papillomas to SCCs. ETS2-deficiency disrupts the SCC-SC super-enhancer landscape and downstream cancer genes while ETS2-overactivation in epidermal-SCs induces hyperproliferation and SCC super-enhancer-associated genes *Fos, Junb* and *Klf5*. Together, our findings unearth an essential regulatory network required for the SCC-SC chromatin landscape and unveil its importance in malignant progression.

## Introduction

Stem cells (SCs) have the capacity to self-renew and to generate and repair tissues. Advances in genome technologies provide the means to not only comprehensively analyze transcriptional profiles of SCs but also map their epigenetic landscape on a global level. Studies on cultured embryonic stem cells (ESCs) have shown that a special set of large open-chromatin domains, so-called super-enhancers (SEs), control expression of genes important to ESC behavior (*Whyte et al., 2013*; *Parker et al., 2013*). Like typical enhancers (TEs), SEs bind with Mediator, a complex that brings promoter and enhancer together to activate transcription. However, SEs differ from TEs by their exceptional size (often >15kb) and by their high density of acetylation of histone H3 at the lysine 27 position (H3K27ac), which characterizes active chromatin.

Studies on a variety of tissue cell lines have shown that another key feature of SEs is a high density of sequence motifs for cell stage-specific transcription factors (TFs). This allows for their cooperative binding, thereby rendering SE-regulated genes particularly sensitive to the key TF cohort. Moreover, the genes encoding lineage-specific TFs often themselves harbor SEs resulting in a stable feed-forward loop to fuel and maintain the lineage (*Whyte et al., 2013*; *Hnisz et al., 2013*; *Lovén et al., 2013*; *Chapuy et al., 2013*).

**eLife digest** Many cancers contain a mixture of different types of cells. Of these, cells known as cancer stem cells can form new tumours and drive the growth and spread of the cancer around the body. A central question is how cancer stem cells differ from healthy adult stem cells. Recent evidence suggests that, in addition to having genetic mutations, cancer stem cells live in a very different environment to other cells within the tumour. This 'microenvironment'also has a major impact on how these cells behave compared to normal stem cells. Together, the genetic and environmental differences profoundly change the way genes are expressed in the cancer cells.

In 2013, a group of researchers identified regions of DNA called super-enhancers. These regions are long stretches of DNA that proteins called transcription factors can interact with to coordinate the expression of nearby genes to alter the production of certain proteins. Super-enhancers contain several transcription factor-binding sites that are close to each other with the different sites being associated with transcription factors that are only active in specific types of cells. Furthermore, super-enhancers are often self-regulatory, meaning that the binding of transcription factors to a super-enhancer can lead to an increase in the expression of the genes that encode the same transcription factors.

Yang, Schramek et al. have now identified the super-enhancers in a skin cancer called squamous cell carcinoma and showed that they differ dramatically from the super-enhancers of normal skin stem cells. Their experiments show that the active super-enhancers in cancer stem cells are associated with a very different set of genes that are highly and often specifically expressed in cancer stem cells. In the cancer stem cells, a transcription factor called ETS2 binds to the super-enhancers and reprograms the expression of genes to promote the development of cancer. Yang, Schramek et al. also show that over-active ETS2 is a major driver of squamous cell carcinoma. Furthermore, ETS2 also increases the expression of genes that cause inflammation and promote the growth of cancers.

Yang, Schramek et al.'s findings reveal a new regulatory network that governs the expression of genes involved in cancer. Furthermore, the experiments show that high levels of ETS2 are linked with poor outcomes for patients with head and neck squamous cell carcinoma, which is one of the most life-threatening cancers world-wide. In the future, these findings might lead to the development of new therapies to treat these cancers.

Mice devote much of their tissue-generating energy to making hair follicles (HFs), each of which possesses a compartment of multipotent SCs that fuel HF regeneration and hair growth. Additionally, hair regeneration is governed by cyclical bouts of SC activity where HFs are either at rest (telogen) or actively regenerating (anagen). The ability to purify large quantities of SCs directly from their native microenvironment ('niche') makes the HF one of the few systems where in vivo chromatin dynamics can be studied in an adult tissue SC population. Such studies on purified HF-SCs show that although their TFs are largely distinct from those of ESCs, many of the basic features of SEs are displayed by HF-SCs in their native tissue niche (*Adam et al., 2015*).

Studies on SC populations isolated from adult tissues have also yielded new insights into SE dynamics. Notably, most of the master regulators of HF-SCs bind within smaller chromatin domains (<3 kb) of SEs termed 'epicenters', which are also enriched for Mediator and for H3K27ac (*Adam et al., 2015*). Additionally, epicenters cloned from HF-SC SEs can drive proper developmental-specific and hair cycling-specific behavior of a GFP reporter, underscoring the functional relevance of these chromatin domains in lineage-specific gene expression.

Perhaps, most intriguing is that even though HF-SCs still possess stemness outside their native niche, for example in culture or in wound-repair, they change their SE landscape dramatically. Mechanistically, this plasticity of remodeling can be traced to a dramatic reduction of HF-SC TFs, resulting in a decommissioning of most in vivo SEs with a new set of SEs gained in vitro (*Adam et al., 2015*). In a related study on tissue macrophages, it was shown that when these immune cells invade different tissues and are faced with new microenvironments, they too display markedly distinct chromatin landscapes (*Gosselin et al., 2014*; *Lavin et al., 2014*). Together, these findings underscore the importance of the microenvironment in governing chromatin dynamics. For SCs, this feature is

particularly relevant, since they reside in specific niches, and receive signals from multiple different surrounding cells to guide their behavior (*Chen et al., 2015*; *Scadden, 2014*; *Lane et al., 2014*; *Morrison and Scadden, 2014*; *Schofield, 1978*).

Cancer can be viewed as a disease of lost cell identity. The cellular origin of tumor-initiating, so-called 'cancer SCs' may be a result of arresting SC maturation or of dedifferentiation of a progenitor lineage, endowing the incipient tumor cell with the potential to self-renew and fuel the cancer. Although largely limited to studies on self-renewing cancer cell lines in vitro and their normal whole tissue (e.g. MCF7 breast cancer cell line and mammary epithelium), increasing evidence underscores the importance of SEs in cancers (*Parker et al., 2013*; *Hnisz et al., 2013*; *Lovén et al., 2013*; *Chapuy et al., 2013*; *Chipumuro et al., 2014*).

Recent studies suggest that the transcriptional profile and behavior of tumor-initiating SCs are highly sensitive to their tumor-microenvironment, which is markedly distinct from that of their normal counterparts (*Plaks et al., 2015*; *Oshimori et al., 2015*). This makes it tempting to speculate that the SE profile of tumor-initiating SCs in vivo may be substantially different from that of normal tissue SCs. If so, SE profiling could provide an avenue for identifying the special set of cancer SC TFs that drive tumor progression. This knowledge could greatly accelerate the convergence of basic science and clinical translation into therapeutics and diagnostics that cripple oncogenic behavior.

Here, we specifically test this hypothesis, focusing on skin squamous cell carcinoma (SCC), which is one of the most common and rapidly rising cancers world-wide (*Rogers et al., 2010*; *Jemal et al., 2010*; *Trakatelli et al., 2007*). More broadly, SCCs can also arise in lung, breast, esophagus, cervix, and oral tissues of the head and neck, where they are associated with high risk of metastasis and mortality (*Trakatelli et al., 2007*). Using SE and transcriptional profiling, we first define the SE chromatin landscape and its gene expression program within the tumor-initiating SC population of skin SCCs and show that SE-associated genes constitute a signature that specifically associates with cancer-vulnerability genes, some of which correlate with poor survival among human SCC patients. Given that HF-SCs are an established origin of these cancers (*Lapouge et al., 2011*; *White et al., 2011*), we conduct comparative analyses and show that SCC-SC SEs are distinct from SEs found in HF-SCs and also from rapidly proliferating short-term progeny of HF-SCs. We show that SCC-SEs are enriched for sequence motifs of a unique set of TF families, whose members are themselves regulated by SEs. Most importantly, we show by gain and loss of function and by associated chromatin landscaping, that these factors are essential for driving the chromatin dynamics that define the malignant state and govern tumor maintenance and survival.

## Results

### The SEs of SCC stem cells in vivo differ dramatically from their normal counterparts

To explore the in vivo importance of SEs in tumor-initiating cancer SCs, we first set up a reliable and reproducible SCC allograft model. The vast majority of chemically induced SCCs in mice have mutations in *Hras, Kras* or Rras2 (*Nassar et al., 2015*), and HRas$^{G12V}$ alone is sufficient to induce formation of benign tumors (papillomas) (*Chen et al., 2009*). HRas$^{G12V}$ in combination with loss of TGFβ receptor II (TGFβRII) results in invasive SCCs, which can metastasize (*Guasch et al., 2007*; *Lu, 2006*; *Bian et al., 2009*). We therefore purified primary keratinocytes from skin of newborn mice harboring a conditional *Tgfbr2* allele (*Tgfbr2$^{fl/fl}$*) encoding a key transmembrane receptor for TGFβ signaling, and infected them with a non-integrating adenovirus expressing a Cre-GFP recombinase. After fluorescence-activated cell sorting (FACS), purified cells were then stably transduced with an integrating retrovirus expressing HRas$^{G12V}$ (strategy outlined in *Figure 1—figure supplement 1A*).

As expected, transduced cells showed increased Ras/MAPK signaling and abrogated TGFβ/SMAD signaling (*Figure 1—figure supplement 1B and C*). Upon intradermal injection into *Nude* mice, they efficiently formed SCC tumors, typified by hyperproliferation, pyknotic nuclei, a discontinuous basement membrane and signs of invasion into the surrounding stroma (*Figure 1—figure supplement 1D*). With this system, tumor-initiation and progression were highly reproducible.

Irrespective of whether chemically or genetically induced, tumor-initiating SCs of SCCs reside at the tumor-stroma interface and are highly enriched for integrins α6 and β1 (*Oshimori et al., 2015*; *Maston et al., 2006*; *Dowen et al., 2014*; *Lapouge et al., 2012*). To profile the SEs of SCC-SCs, we

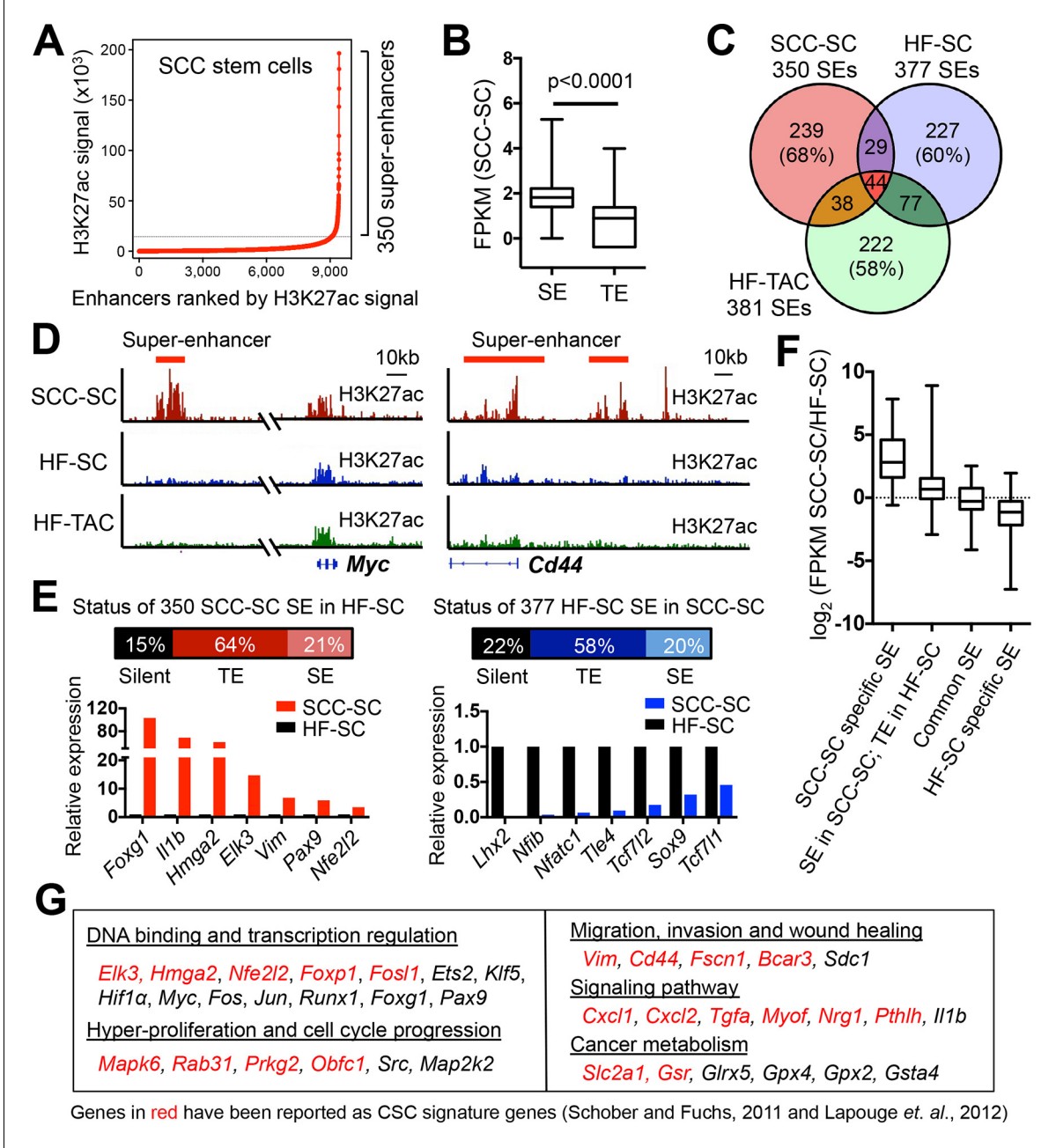

**Figure 1.** Chromatin mapping reveals striking differences between normal skin SCs and SCC-SCs. (A) Identification of H3K27ac super-enhancers in SCC-SCs. (B) Differential gene expression levels driven by SCC-SC super-enhancers and typical-enhancers. p-Values are from t-test. (C) Venn diagram showing that super-enhancers of SCC-SCs show little overlap with HF-SCs or HF-TACs. (D) Examples of SCC-SC-specific super-enhancers at *Myc* and *Cd44* loci. (E) Differences between HF-SC and SCC-SC super-enhancers. Note the decommissioning of HF-SC master regulators in SCC-SCs and corresponding suppression of HF-SC TF expression. (F) Enhancer remodeling correlates with gene expression changes. Boxplot displaying the full range of gene expression changes (min. to max.). (G) Selected genes associated with SCC-SC super-enhancers. HF, hair follicle; SC, stem cell; SCC, squamous cell carcinoma; TF, transcription factor.

The following figure supplements are available for figure 1:

**Figure supplement 1.** Validation of the allograft tumor model.

**Figure supplement 2.** Super-enhancer profiling in SCC-SCs.

therefore employed FACS to purify the GFP$^{high}$α6-integrin$^{high}$β1-integrin$^{high}$ population from *Tgfbr2*-null HRas$^{G12V}$-oncogenic skin SCCs (*Figure 1—figure supplement 1E*) (*Schober and Fuchs, 2011*). We then took advantage of the fact that when chromatin immunoprecipitation and high-throughput sequencing (ChIP-seq) is carried out for Mediator 1, H3K4me1, and H3K27ac, they generate highly overlapping patterns within SEs (*Whyte et al., 2013*; *Adam et al., 2015*). Using H3K27ac as our paradigm, we performed ChIP-seq and delineated the SEs of our purified SCC-SCs. Independent biological replicates showed a high degree of similarity (Pearson correlation coefficient of genome-wide read densities r>0.89; representative example in *Figure 1—figure supplement 2A*), so the replicate data were subsequently combined and processed to maximize the comprehensiveness of our SE analysis.

H3K27ac peaks resided within promoters (± 2 kb of annotated genes) (43%) and distal elements, considered enhancers (57%) (*Figure 1—figure supplement 2B*). A small fraction (13%) of all enhancers (*Figure 1—figure supplement 2C*), were in close proximity (<12.5 kb apart) and highly enriched with H3K27ac; they formed ~350 unusually large (median size >20 kb) distal elements, thereby fulfilling the criteria for 'SEs' (*Whyte et al., 2013*) (*Figure 1A*, *Figure 1—figure supplement 2C–E*). SEs were also significantly more robust than TEs in their ability to drive gene expression in SCC-SCs (*Figure 1B*).

When compared to the SEs of HF-SCs (*Adam et al., 2015*), that is well-established precursors for skin SCCs (*Lapouge et al., 2011*; *White et al., 2011*), it was readily apparent that the SE landscape had been dynamically remodeled in SCC-SCs. This was not attributable merely to the difference in proliferative status, as the SE landscape of SCC-SCs was also distinct from that of rapidly proliferative, short-lived HF-SC progeny (transit-amplifying cells, TACs) (*Figure 1C*).

## SCC-SC SEs associate with genes that are highly upregulated in cancer

Enhancers control adjacent genes by looping to their promoters, with most of these interactions occurring within <50 kb (*Maston et al., 2006*). More than 80% of SEs can be accurately assigned to their target genes by applying proximity criterion and RNA-seq expression data (*Dowen et al., 2014*). Most of the remaining ambiguities arise from situations where more than one active gene resides within the vicinity of SEs for a particular cell type (*Dowen et al., 2014*). These can largely be resolved by extending comparative ChIP-seq and RNA-seq analyses to multiple lineage stages for a particular cell type (*Adam et al., 2015*). Therefore, after conducting RNA-seq analysis on the GFP$^{high}$α6$^{high}$β1$^{high}$ SCC-SC population, we assigned SE-associated genes on the basis of 1) their proximity to an SCC-SC SE; 2) their active transcription in SCC-SCs; and 3) their strict correlation between expression pattern and the presence of their putative SE not only in SCC-SCs but also in HF-SCs and transit-amplifying progeny.

On the basis of this analysis, we readily assigned 340 genes to the 350 SEs found in SCC-SC chromatin, as 10 genes appeared to be controlled by more than one SE (*Supplementary file 1*). The cohort of SE-associated genes included a number of oncogenes such as *Myc*, and also genes associated with SCCs, for example *Cd44* (*Figure 1D*). Overall, 15% of the SCC-SEs completely lacked H3K27ac signals in HF-SCs, while 22% of the HF-SEs were silenced in SCC-SCs (*Figure 1E*). Notably, the master transcriptional regulators of HF-SCs were largely decommissioned in SCC-SCs, while new transcriptional regulators were activated. Based on RNA-seq analysis, the genes associated with SCC-specific SEs displayed the highest increases in expression between HF and SCC-SCs, while genes associated with HF-specific SEs showed the greatest decline in expression in SCC-SCs (*Figure 1F*).

Sixty-four per cent of SCC SE-associated genes were still expressed in HF-SCs but had lost their SE and acquired a TE. In many cases, these enhancer dynamics had significant consequences, since the overall expression levels (FPKM) of SE-associated genes were higher than those of TE-associated genes (*Figures 1E and F*), in agreement with previous findings for cultured ESCs (*Whyte et al., 2013*). By contrast, genes which were associated with SEs both in SCC-SCs and HF-SCs were expressed at comparable levels (*Figure 1F*).

Unbiased gene ontology (GO) analysis categorized SE-associated genes in SCC-SCs as wound-responsive, stress-responsive, TF binding, kinase targets or actin binding, according to molecular function and biological process (*Figure 1—figure supplement 2F and G*). *Myc* was particularly interesting, in that human *MYC* has been shown to be associated with an SE in a variety of cultured cancer cell lines (*Hnisz et al., 2013*). In addition to *Myc*, there were a number of other established

oncogenes that had SE specifically in SCC-SCs but not in HF-SCs, including *Fos, Jun, Src,* and *Tgfa.* Also on this list, there were cytokine genes *Cxcl1* and *Cxcl2*, as well as genes associated with cancer metabolism such as *Slc2a1* and *Gsr* (*Figure 1G*). Many of these genes also scored as ≥2X up-regulated in purified SCC-SCs relative to their normal counterparts in either epidermis or HF (*Schober and Fuchs, 2011*; *Lapouge et al., 2012*). These findings underscored the importance of SE-regulated genes in cancer.

## Unraveling the network of master transcriptional regulators of SCC stem cells in vivo

Next, we sought to identify the key TFs involved in regulating the SE landscape in SCC-SCs. An unbiased motif analysis of SCC-SC SEs revealed a distinct set of putative TF binding sites that were largely non-overlapping with those found in the SEs of HF-SCs and TACs (*Adam et al., 2015*) (*Figure 2A*). ETS was the most frequently found motif (~80%), with SOX and AP1 motifs found in >70% of all SCC-SEs (*Figure 2B*). Notably, these putative binding sites occurred within epicenters, that is, small regions (1.5–3 kb) of SE chromatin that were particularly enriched for the H3K27ac mark. Moreover, among SEs with both ETS and AP-1 sequence elements, ~40% contained the two motifs within a 100bp stretch, meeting conditions for potential cooperative binding.

Prior studies on SEs suggest that the key regulatory TFs are those whose genes are regulated by SEs ( *Whyte et al., 2013*; *Adam et al., 2015*). Among the SE-regulated genes were three AP-1 motif genes (*Fos, Junb, Nfe2l2*), three KLF motif genes (*Klf5, Sp2, Sp3*), two ETS motif genes (Elk3, Ets2), and Hoxb7 (*Figure 2C*). SOX2 is a well-established SCC-TF (*Liu et al., 2013*; *Siegle et al., 2014*; *Boumahdi et al., 2014*), although its gene had a TE and not an SE. When possible, immunofluorescence confirmed their expression (*Figure 2D*).

Functional analyses previously highlighted the importance of *Fos, Junb, Nfe2l2,* and *Sox2* in SCCs (*Oshimori et al., 2015*; *Liu et al., 2013*; *Siegle et al., 2014*; *Boumahdi et al., 2014*; *Gao et al., 2014*; *Briso et al., 2013*; *Ding et al., 2013*). However, with the exception of a report of an elevation in *Ets2* in ~75% of esophageal cancer patients (*Li, 2003*), an association of ETS proteins and SCCs has not been hitherto described, and functional analyses in SCCs are entirely lacking for this family.

Both ELK3 and ETS2 were expressed and nuclear in our skin SCCs (*Figure 2D*). For ETS2, we could even confirm that this TF was threonine 72 (T72) phosphorylated, that is active, in SCC-SCs (shown). This modification is known to be dependent on Ras-MAPK kinase and results in an increase of its DNA-binding affinity and interaction with the histone acetylation complex CBP/p300 (*Foulds et al., 2004*). Interestingly, active phosphorylated ETS2 was also readily detected in human SCCs, as judged by immunohistochemistry (*Figure 2E*).

Finally, and most importantly, when we surveyed the TCGA database and analyzed the relation between *Elk3* and *Ets2* expression versus median survival of SCC patients, a clear correlation was observed between the level of expression of these genes individually and poor prognosis (*Figures 2F and G*). For patients with high *Elk3* and/or *Ets2* expression, the median survival was ~3–6 fold less than that of patients with lower expression levels. Based on these analyses, ELK3 and ETS2 became prime candidates to participate in autoregulation of the SCC-SC SEs.

## SE-associated ETS TFs constitute cancer vulnerabilities within SCCs

Our ChIP-seq analyses showed that genes encoding both ELK3 and ETS2 were governed by SEs. To test their functional relevance, we began by focusing on ELK3 as its expression was quantitatively suppressed in epidermal (Epi) progenitors and quiescent HF-SCs, and was only transcribed at low levels in proliferative progenitors during the growth phase of the hair cycle. By contrast, *Elk3* was robustly transcribed in SCC-SCs, where its encoded protein ELK3 was nuclear (*Figures 1E*, *3A* and *3B*).

To assess the physiological relevance of ELK3 in SCC tumorigenesis, we identified three distinct shRNA hairpins that knocked down *Elk3* mRNA and protein levels by >70% (*Figure 3C*). We transduced the same starting population of SCC-SCs with each of these shRNAs and with a control scrambled (SCR) shRNA and tested their abilities to affect tumor growth when injected intradermally into host recipient mice.

Relative to scrambled (SCR) control shRNAs, each *Elk3* shRNA efficiently knocked down the expression of ELK3 protein in the resulting tumor tissue, and exerted a potent effect, with signs of

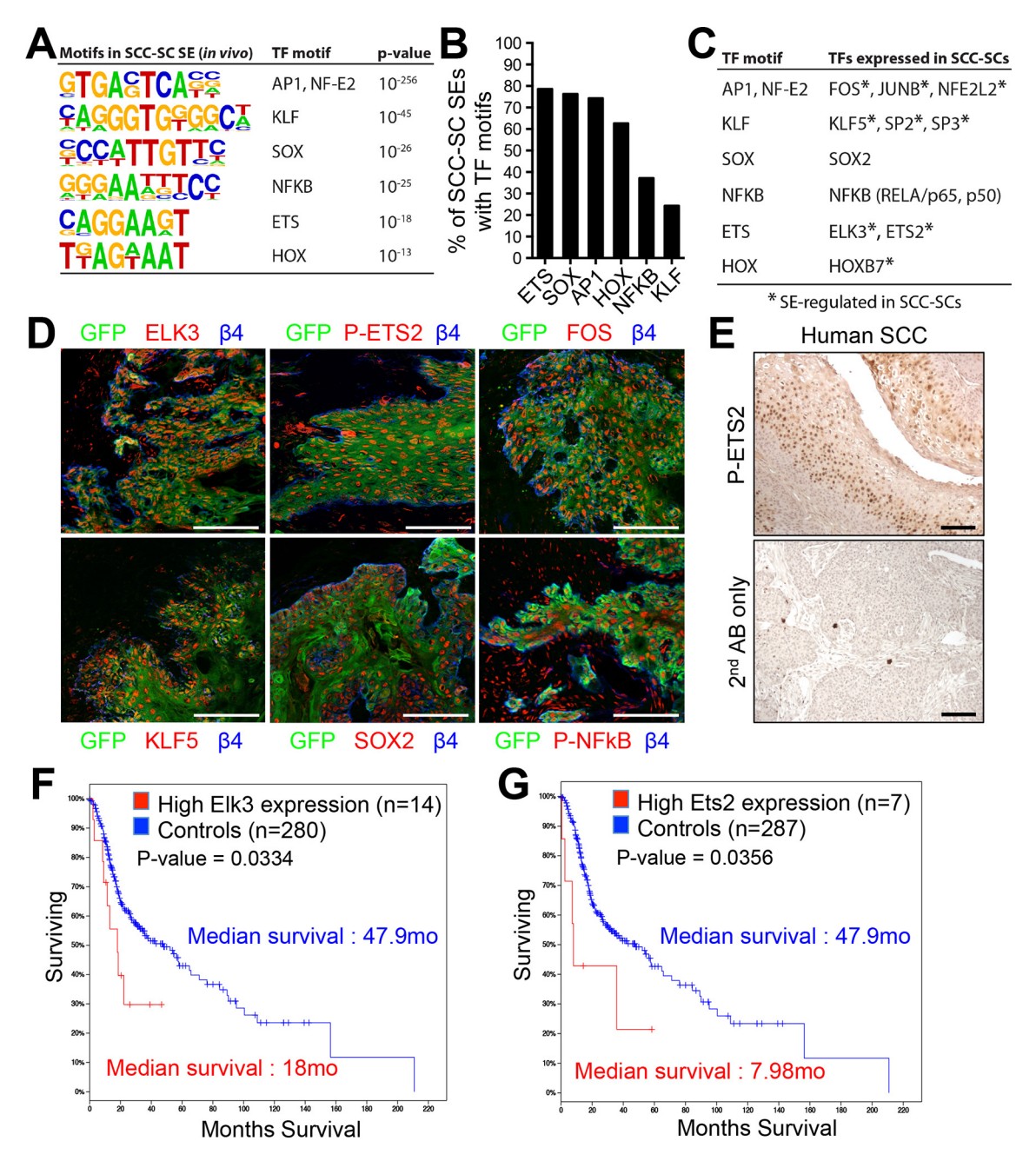

**Figure 2.** Identification of a cohort of SCC-SC specific transcriptional regulators. (**A**) Motif analysis of SCC-SC super-enhancers for putative TF binding sites. (**B**) Frequency of putative TF binding sites in SCC-SC super-enhancers. (**C**) SCC-SC TFs with potential to bind to the TF-motifs within SCC-SC super-enhancers. Genes encoding the TFs that are marked with an asterisk are SE-associated. (**D**) Immunofluorescence images showing nuclear localization of SCC-SC TFs (red) in allograft SCC-SC-derived tumors (GFP). Scale bars, 100 μm. (**E**) Immunohistochemistry with P-ETS2 antibodies in human SCC samples. (**F and G**) High expression of mouse SCC-SC-expressed ETS family members correlate with poor survival in human SCCs. Kaplan–Meier analysis compares overall survival of TCGA head and neck SCC patients stratified according to high and low ELK3 and ETS2 expression. SC, stem cell; SCC, squamous cell carcinoma; TF, transcription factor;

reduced allograft tumor growth appearing as early as 8–9 days after injection (*Figure 3D and E*). Moreover, the morphologies and corresponding biomarkers of resulting cell masses were also dramatically changed in ELK3-deficient tumors. As shown in *Figure 3F*, SCC-SCs transduced with scrambled (SCR) control viruses generated tumors which were undifferentiated and almost exclusively

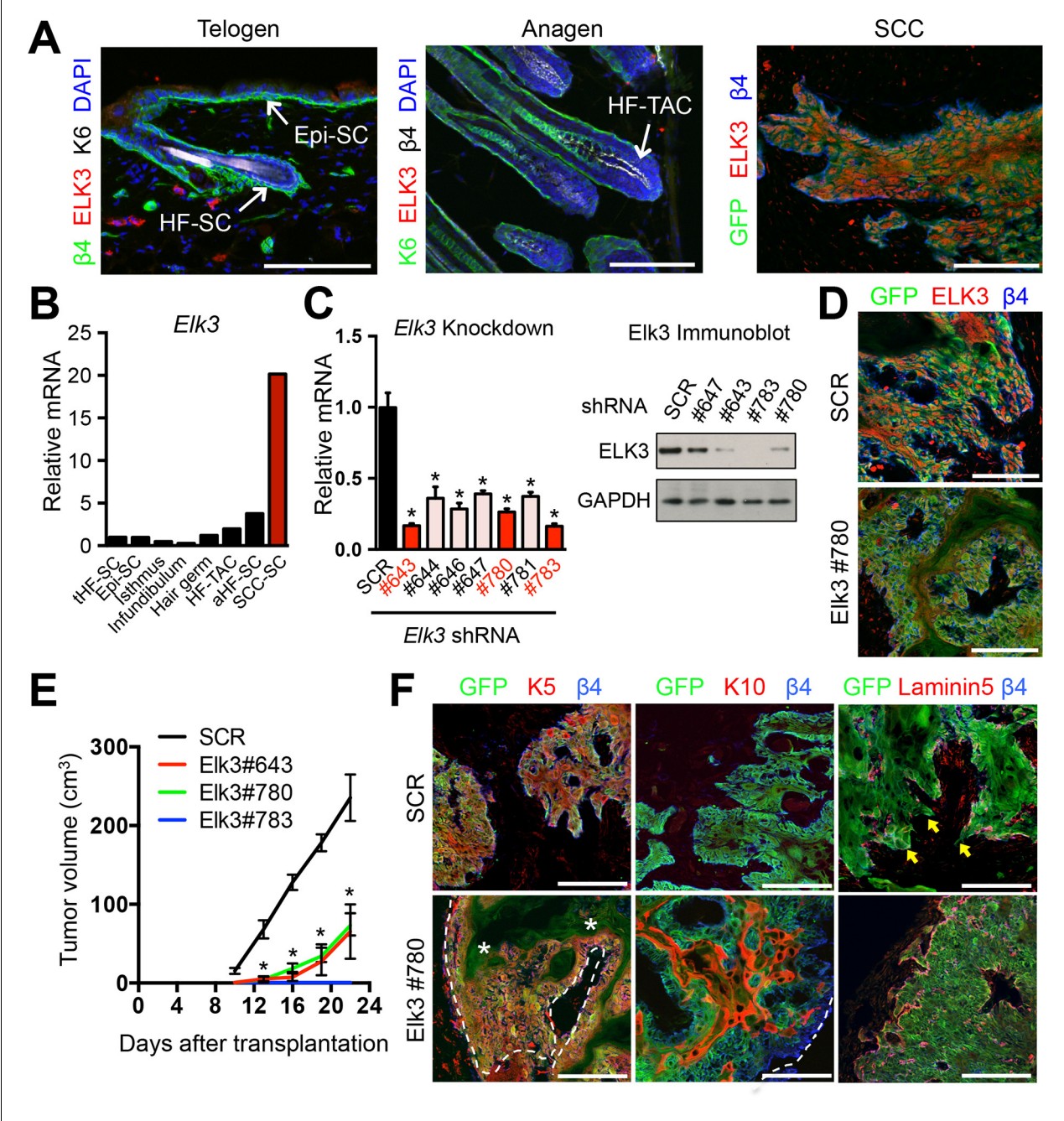

**Figure 3.** ELK3 is specifically induced in SCC-SCs and promotes SCC growth. (A) Immunofluorescence analysis showing that ELK3 is absent in normal skin epithelia but expressed in SCC-SC-derived allograft tumors. Epi-SC, epidermal basal cells; HF-SC, hair follicle stem cells; HF-TAC, short-lived, transit-amplifying cells. Scale bars, 100 μm. (B) *Elk3* gene expression is low or absent in normal skin cells but specifically induced in SCC-SCs. Telogen(t) and anagen(a) HF-SCs; isthmus, infundibulum, and hair germ are other progenitor compartments of the HF. (C) (left) Knockdown efficiency of *Elk3* shRNAs in *HRas*$^{G12V}$; *Tgfbr2*-null cells as measured by quantitative RT-PCR. (n = 3 ± SEM; *p<0.05). (right) Immunoblot analyses confirming that changes in mRNA levels are reflected at the level of protein. (D) Immunofluorescence image of allograft tumors (GFP) from control and *Elk3*-knockdown SCC-SCs. Scale bars, 100 μm. (E) Changes in tumor volume of *Elk3* knock-down and control allografts over time (n = 3). (F) Immunofluorescence images of *Elk3* knock-down allograft tumor (GFP) showing reduced numbers of undifferentiated keratin 5 (K5) cells, appearance of keratinized pearls of cells expressing differentiation marker keratin 10 (K10), and diminished signs of basement membrane breakdown (as judged by laminin 5) and of invasion at the tumor-stromal interface. Asterisks mark absence of K5 in keratinized pearls. Dotted line marks tumor-stromal boundaries. Arrows denote invasive tumor cells at signs of discontinuous basement membrane. Scale bars, 100 μm. HF, hair follicle; SC, stem cell; SCC squamous cell carcinoma.

positive for keratin-5 (K5), a marker of cells with proliferative progenitor potential (*Fuchs and Green, 1980*; *Blanpain and Fuchs, 2014*). By contrast, ELK3-deficient SCC-SCs formed tumors with pearls of keratinized K5-negative cells. These pearls were positive for keratin-10 (K10), a classic marker of terminally differentiating cells. Moreover, in contrast to classical signs of SCCs, including invasion and discontinuous basement membrane, tumors derived from ELK3-deficient SCC-SCs displayed smooth borders and quite continuous anti-laminin 5 immunolabeling at the tumor-stromal border (*Figure 3F*).

We next turned to ETS2. Although active (phosphorylated) ETS2 protein was only seen in SCCs and not in normal progenitors, the *Ets2* gene was more broadly and abundantly transcribed in pro-genitors, unlike *Elk3*. Based on these data, it was equally important to examine the consequences of *Ets2* loss of function on SCC tumorigenesis. To do so, we selected two powerful hairpins, which knocked down *Ets2* transcription and protein production by >65% (*Figure 4A*). Importantly, trans-duction of SCC-SCs with *Ets2*-shRNA also resulted in a loss of ETS2 protein as judged by immuno-labeling of tumors derived from injecting these cells into host recipient mice (*Figure 4B*). Of additional note, even though ETS2 was detected in the stromal cells, ETS2 protein was selectively absent in the *Ets2*-shRNA-transduced tumor masses. Together, these findings confirmed the efficacy of the knockdown.

Whether the SCC cells were transduced with *Ets2* #649 or #985, tumor growth was markedly reduced relative to control virus (*Figure 4C*). Similar to the effects of *Elk3*-shRNAs, the morphologies and corresponding K5/K10 biomarkers of the resulting cell masses were also dramatically changed in ETS2-deficient tumors. Additionally, throughout much of the ETS2-deficient tissue, the β4 integrin and laminin5 patterns were uncharacteristically continuous, indicative of a well-organized underlying

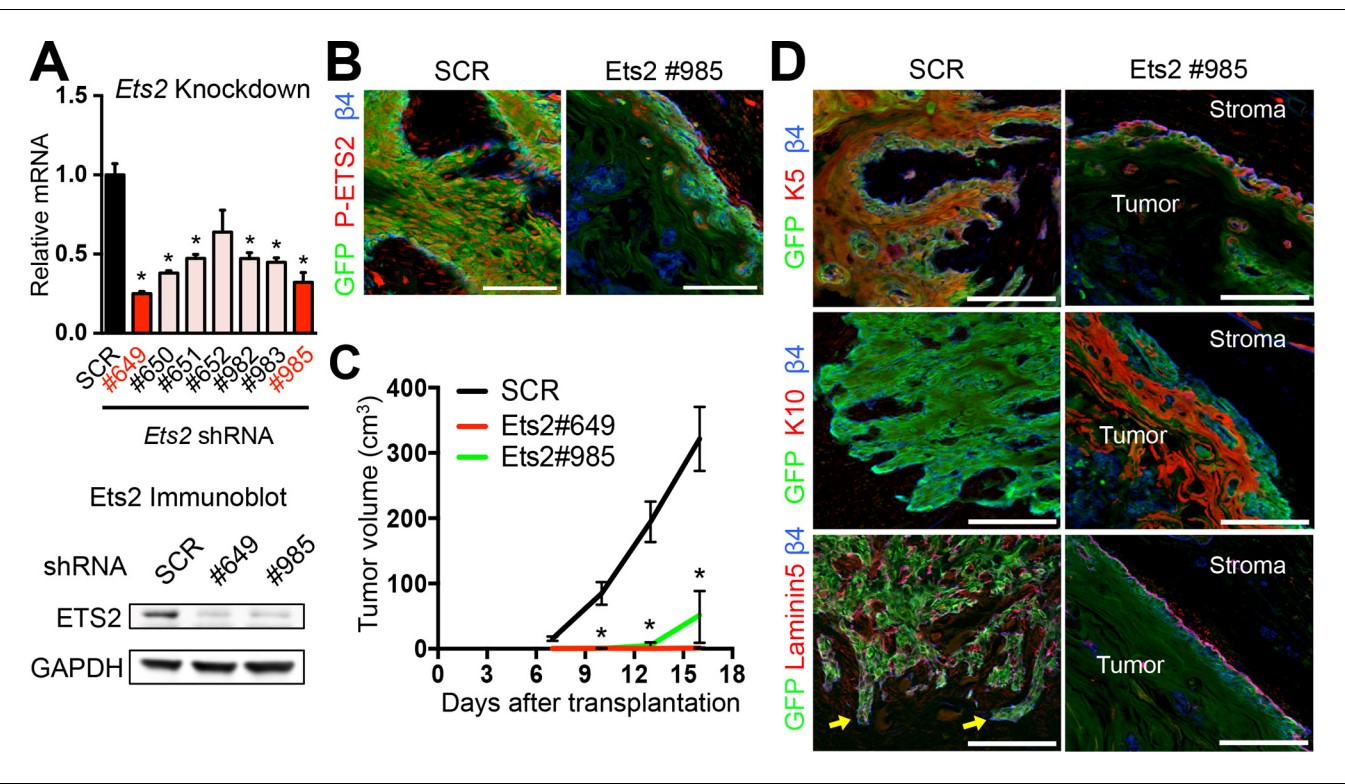

**Figure 4.** ETS2 governs SCC growth and malignancy. (A) (left) Knockdown efficiency of *Ets2* shRNAs in *HRas^{G12V}*; *Tgfbr2*-null cells as measured by quantitative RT-PCR. (n = 3 ± SEM *p<0.05). (right) Immunoblot analyses confirming that changes in mRNA levels are reflected at the level of protein. (B) Immunofluorescence image of allograft tumor (GFP) showing reduced ETS2 protein in *Ets2* knock-down tumors. Scale bars, 100 µm. (C) Changes in tumor volume of *Ets2* knock-down and control allografts over time (n = 3). (D) Immunofluorescence images of allograft tumors (GFP) of transduced SCC-SCs. Note reduction in undifferentiated K5+ cells and appearance of keratinized pearls of K10+ differentiated cells upon *Ets2* knock-down. Of note, yellow arrows depict regions of discontinuous laminin5 and integrin β4 staining, indicative of a disrupted basement membrane and local tumor invasion. Scale bars, 100 µm. SC, stem cell; SCC, squamous cell carcinoma; SEM, standard error of the mean.

basement membrane. Concomitantly, very few finger-like projections into the surrounding stroma were seen with either *Ets2*-shRNA hairpins. Thus, while the tissue derived from *Ets2* knockdown SCC cells was still disorganized, many of the classic signs of SCCs were lost without this SE-associated TF. Conversely, morphological and biochemical features of benign papillomas were enhanced in HRas$^{G12V}$; *Tgfbr2*-null tumor masses in the absence of ETS2 (*Figure 4D*).

## The shRNA-mediated effects on SCCs depend upon ETS levels, which also impact other TFs with SE binding motifs

Overall, the impeding effects on SCC tumor growth and morphology were seen irrespective of whether we knocked down *Ets2* or *Elk3*. To unequivocally document the tumor-promoting effects of ETS proteins in skin, we performed rescue experiments with an shRNA-resistant *Ets2*-cDNA (*Figure 5—figure supplement 1A*). When SCC cells were transduced with both the *Ets2*-shRNA and the *Ets2*-cDNA harboring a mutated sequence in the shRNA targeting site (*Ets2_mut*) and then injected subcutaneously into host recipient mice, the *Ets2_mut* refractory to shRNA knockdown rescued the effects of *Ets2*-shRNA. As shown in *Figure 5A*, SCC tumor growth and morphology was quantitatively restored, indicating that the effects on SCCs were directly rooted in the relative levels of ETS expression.

Previously, we showed that for HF-SCs, sustained levels of SOX9 were critical for the binding of other HF-SC TFs to SE epicenters (*Adam et al., 2015*). Given the importance of ETS proteins, we wondered what the consequences of diminishing their expression might be on the SCC-SC TFs harboring binding motifs within the SCC-SEs. Interestingly, as shown in *Figure 5B*, loss of ETS2 abrogated expression of SOX2, KLF5, and ELK3. In this regard, ETS2 was to SCC-SC SEs what SOX9 was to HF-SC SEs.

If ETS factors function in SE dynamics, then elevating the levels of active ETS factors in normal skin progenitors might be expected to induce some of the phenotypic consequences of SCC progression, including hyperproliferation and the expression of other SCC-TFs or SCC genes that are regulated by SEs. To test this hypothesis, we engineered a phosphomimetic T72D version of ETS2, thus bypassing the need for HRas/MAPK to make this modification (*Foulds et al., 2004*; *O'Neill et al., 1994*; *Brunner et al., 1994*; *Rebay and Rubin, 1995*). Since ETS2 is also expressed in normal skin progenitors, we added a Myc-tag to monitor protein expression. Finally, we used a tetracycline regulatory element (TRE) to induce protein expression, and packaged the transgene in a lentivirus that also harbored a constitutively active H2B-RFP. When tested in transduced primary keratinocytes in vitro, expression of phosphomimetic T72D and wild-type ETS2 was markedly upregulated upon Doxycycline administration, where proteins, expressed at comparable levels, could be readily detected by ETS2 and Myc-antibodies (*Figure 5—figure supplement 1B*).

With these controls, we then proceeded to inject the high titer lentivirus into the amniotic sacs of E9.5 K14-rtTA mouse embryos in utero. As previously shown, this strategy selectively and efficiently transduces the single layer of surface epithelial progenitors of the embryo, particularly within the head region (*Beronja et al., 2010*). Within 24 hrs, the lentiviral targeted DNA stably integrates into the host genome and can be stably propagated into the adult skin. By adding Doxycycline to the mouse chow, we could control the activation of the rtTA transactivator (*Nguyen et al., 2006*), and hence the timing at which the TRE-T72D-ETS2 protein was expressed. *Figure 5C* illustrates this strategy. As shown in *Figure 5D*, Doxycycline induced overexpression of myc-tagged ETS2 transgene was confirmed.

In vivo mice in which overexpression of wild-type ETS2 was induced in the skin epithelium did not exhibit overt phenotypic perturbations, and by immunofluorescence, epidermal differentiation appeared normal (*Figure 5—figure supplement 1C*). By contrast, when the activated form (T72D) of ETS2 was induced, marked epidermal thickenings and invaginations were observed, accompanied by elevated immunolabeling for Ki67, a marker of actively proliferating cells (*Figure 5E* and *Figure 5—figure supplement 1D*). Additionally, immunofluorescence for the endothelial marker CD31 revealed signs of enhanced angiogenesis and FACS for CD11b suggested an increase in inflammatory cells (*Figure 5—figure supplement 1D and E*). All these features are hallmarks of SCCs.

Notably, T72D-ETS2 activated epithelium was also typified by expression of AP1 factors FOS and JUNB, as well as ELK3 (*Figure 5F*). Intriguingly, KLF5, whose gene is also SE-associated in SCCs, was also markedly upregulated, even though it is expressed at lower levels in normal epidermis (*Figure 5F*).

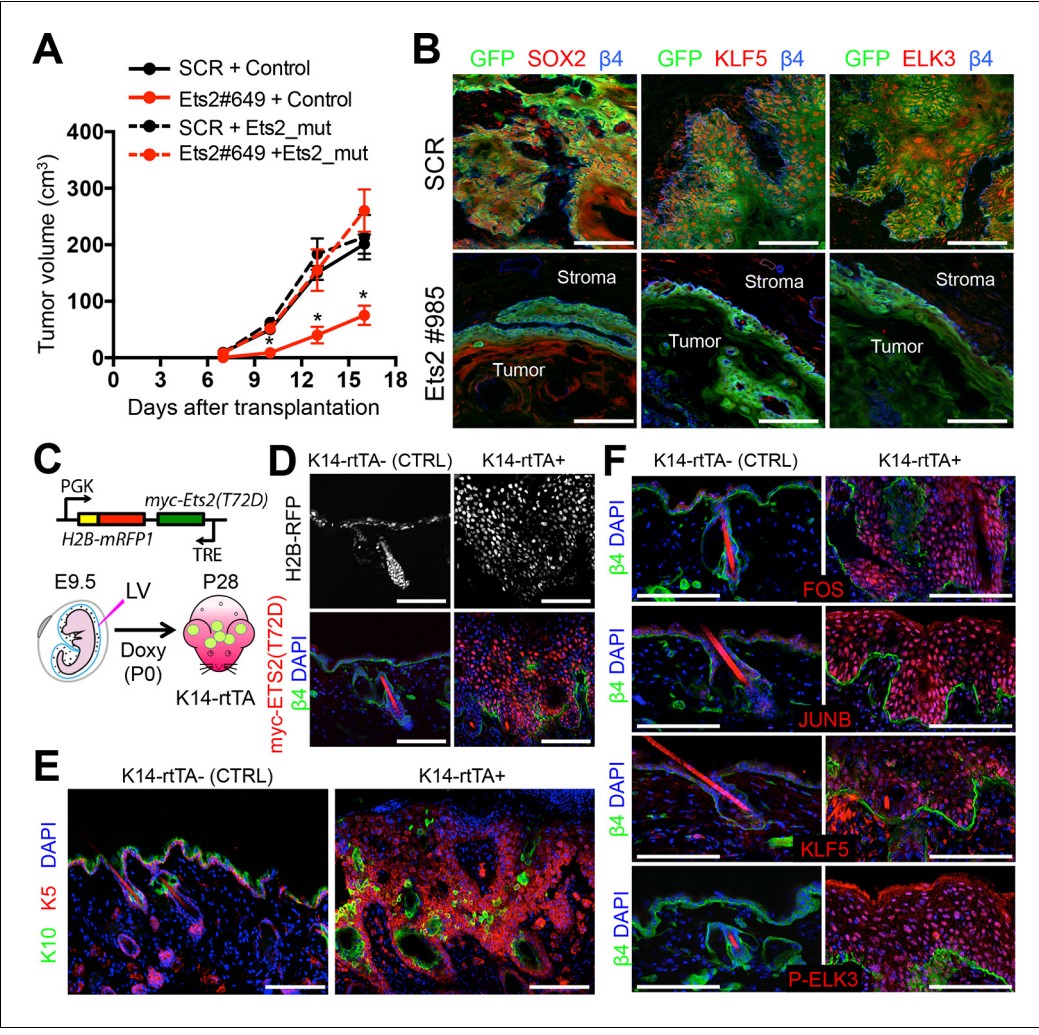

**Figure 5.** ETS2 controls a transcriptional network driving SCC growth. (**A**) Rescue of SCC-SC growth by expressing an *Ets2* cDNA harboring silent mutations in the *Ets2*-shRNA target site. (**B**) Immunofluorescence images of allograft tumors (GFP) derived from transduced SCC-SC cells. Note loss of SOX2, KLF5, and ELK3 expression in *Ets2* knock-down but not control tumors. Scale bars, 100 μm. (**C**) Schematic of strategy to induce expression of a constitutively active, ETS2 (T72D) in normal skin. (**D**) Validation of efficient *in utero* transduction (H2B-RFP) and postnatal ETS2-T72D Doxy-induction in skin epithelium. (**E**) ETS2-T72D expression induces expansion of undifferentiated K5+ cells resulting in epidermal thickening and invagination. (**F**) Induction of constitutively active ETS2-T72D in epidermal progenitors results in marked upregulation of four additional TFs which have sequence motifs in >70% of SCC-SC super-enhancers and whose genes are themselves regulated by super-enhancers. SC, stem cell; SCC, squamous cell carcinoma.

The following figure supplement is available for figure 5:

**Figure supplement 1.** Validation of ETS2 expression constructs.

Overall, these changes were markedly distinct from *Ets2* knockout mice, whose epidermal homeostasis appeared normal (*Yamamoto et al., 1998*). When taken together, our findings suggest a specific role for ETS-family members in regulating skin tumor growth and malignant progression.

## Testing the physiological significance of ETS2 in SE dynamics

While our findings thus far underscored the importance of SEs and ETS proteins, it was also important to test the functional relevance of SE dynamics. To do so, we first determined the transcriptional consequences of selectively inducing constitutively active ETS2 in the skin epithelium of postnatal

mice. Interestingly, T72D-ETS2 not only generated SCC-like hyperproliferation and invaginations, but also induced and/or up-regulated many genes which we had found to be strongly expressed in SCCs (*Figures 6A and B*). Among the genes upregulated by ≥2X upon ETS super-activation in normal skin progenitors (EpiSCs), nearly 50% were also enhanced in SCC-SCs. *Cxcl1* (>300X), *Cxcl2* (>50X), and *Elk3* (4X) ranked among the most differentially expressed.

Notably, many of the genes that were markedly up-regulated in T72D-ETS2 expressing skin epithelium also displayed SE in SCCs (highlighted in red in *Figure 6A*). To address whether these T72D-ETS2 driven transcriptional dynamics were a reflection of SCC-like changes in the chromatin landscape, we purified α6$^+$Sca1$^+$ basal progenitors from the skin epidermis of our mice at 4 weeks after ETS induction, and then subjected these and control epidermal cells to ChiP-seq analysis for the SE mark H3K27ac. Remarkably, 46% of the SEs found in SCC-SCs overlapped with those of T72D-ETS overexpressing skin epidermis with a genome-wide Pearson Correlation Coefficient of 0.72 (*Figures 6C–E*). A few genes, such as *Cdh1* and *Neat1* possessed SEs in SCC-SC, T72D-ETS2-EpiSC and EpiSCs, reflective of their sustained high expression in normal and tumor tissue (*Figure 6F*).

To test for direct binding of ETS2 to these SCC-SEs, we performed in vivo ETS2-CHIP-qPCR on induced T72D-ETS SCC-SCs. Using ChIP immunoprecipitations with antibodies against both endogenous ETS2 and also the Myc-tag, we observed clear enrichment of representative SCC-SE epicenters that harbor ETS2 motifs (*Figure 6G*). Together, these findings provide compelling evidence that over-expressing a constitutively activated form of ETS2 bypassed the need for oncogenic Ras-driven MAP kinase activity in eliciting many of the chromatin changes associated with SCCs.

## CXCL1/2 are regulated by SE and ETS2 and play a critical role in SCC formation

To this end, we focused on the acetylation of lysine 27 of histone H3, which renders SE chromatin mutually exclusive for Polycomb (PcG)-mediated repression, typified by a trimethylation mark at this same residue. In this regard, it was intriguing that a small number of genes, including *Cxcl1/2*, *Hmga2*, *Igf2bp2*, and *Vim*, were PcG-repressed in normal skin progenitors (*Lien et al., 2011*), but they displayed H3K27ac-associated SEs in SCC-SCs (*Figure 7A*). Correspondingly, their expression levels were also markedly increased compared to either epidermal progenitors (EpiSCs) or HFSCs (*Figure 7B*).

We were particularly intrigued by the *Cxcl1* and *Cxcl2* locus, as the encoded *Cxcl1* and *Cxcl2* cytokines can act in paracrine fashion to recruit neutrophils and activate angiogenesis, features which promote tumor progression (*Acharyya et al., 2012*) and which are characteristic of the SCC-SC niche, but not that of their normal counterparts (*Oshimori et al., 2015*). Interestingly however, *Cxcr2*, the gene encoding the cognate receptor for CXCL1 and CXCL2, was also elevated in SCC-SCs relative to either HFSCs or EpiSCs (*Figure 7B*). While induction of phospho (active) ERK was sustained for up to 4 hr, activation of AKTand p38 was transient indicative of feed-back regulation, which is known to occur for this pathway (*Avraham and Yarden, 2011*; undefined). This suggested that in addition to its paracrine role, these cytokines may also serve an autocrine role in SCC behavior.

In agreement with this notion, CXCL2 stimulation of HRas$^{G12V}$ transformed, *Tgfbr2*-null keratinocytes resulted in a marked activation of MAPK signaling as typified by increased ERK, AKT, and p38 phosphorylation (*Figure 7C* and *Figure 7—figure supplement 1A*). Thus, CXCR2 regulation offered a good focus for our testing the physiological relevance of SE dynamics in SCC progression and the importance of changes in the niche microenvironment.

Consistent with the fact that the CXCL1/2 locus gained a SE not only in CSC but also upon expression of a constitutively active ETS (*Figure 6A*), we found ETS binding motifs within the SEs of the *Cxcl1/Cxcl2* locus (*Figure 7—figure supplement 1B*) and showed by ETS ChIP-qPCR its binding (*Figure 6F*). To verify the functional significance, we knocked down *Ets2* with our two different shRNAs and examined the consequences to *Cxcl1* and *Cxcl2* expression. As shown in *Figure 7D*, a significant reduction (≥2X) of *Cxcl1* and *Cxcl2* was seen upon transduction with *Ets2* shRNA relative to the *scrambled* control shRNA.

To further test a potential autocrine role of CXCL1 and CXCL2 in SCC progression, we knocked down *Cxcr2* in HRas$^{G12V}$*Tgfbr2*-null keratinocytes and injected the scrambled shRNA-transduced and *Cxcr2*-shRNA transduced cells into host recipient mice to form allografts. As shown in *Figure 7E*

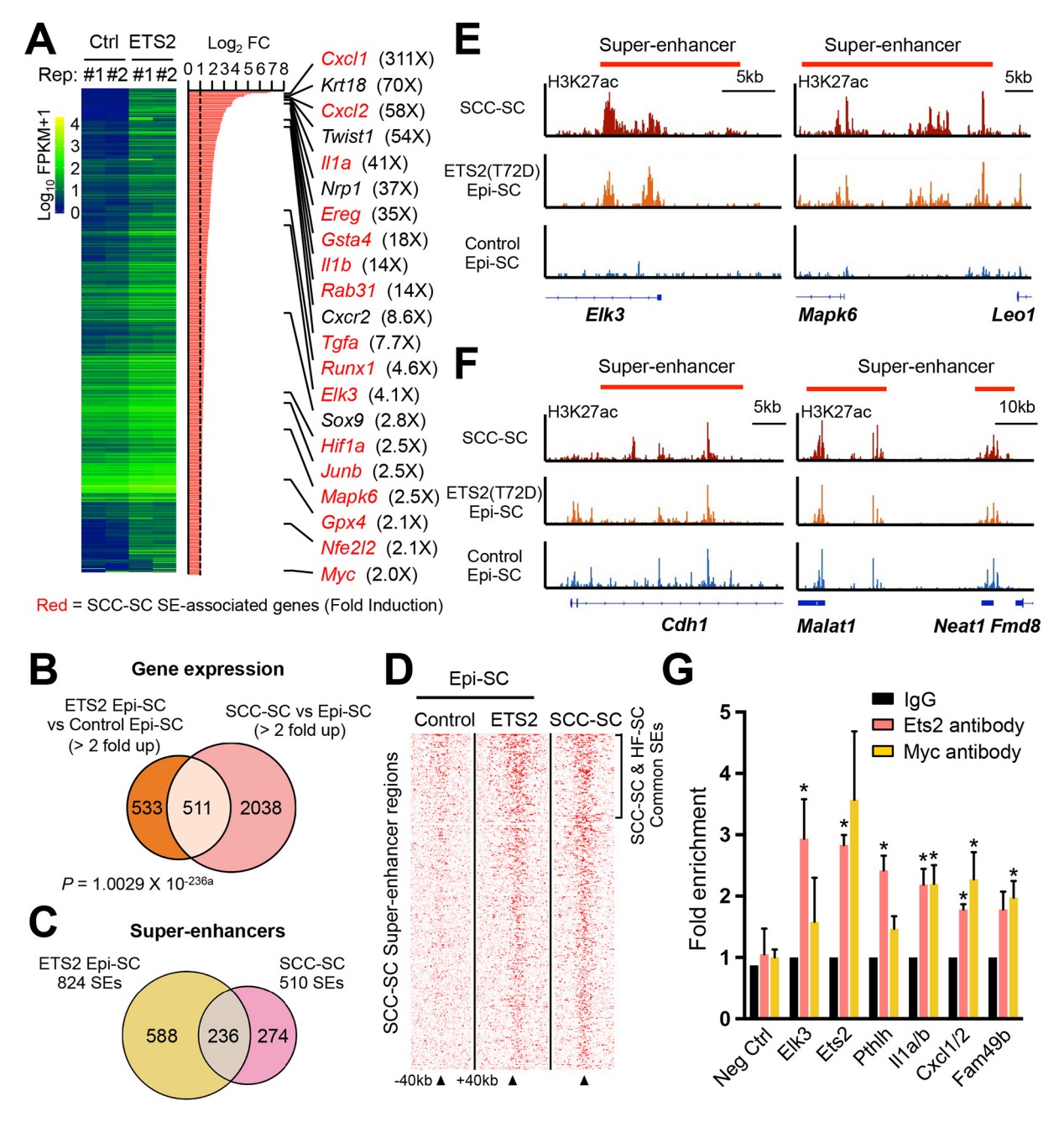

**Figure 6.** Super-activated ETS2 drives chromatin dynamics and transcriptional changes that occur during malignant transformation. (A) Summary of transcriptional profiling of basal epidermal progenitors (Epi-SC) purified from ETS2 (T72D) induced or control skin. Significantly upregulated genes (greater than twofold) were ranked and are listed at right with fold changes. Of note, SCC-SC super-enhancer (SE)-associated genes are marked in red. (B) Venn diagram showing significant overlap between differentially regulated transcripts in T72D-ETS2 Epi-SCs and SCC-SC as compared to Epi-SC. (C) Venn diagram showing that SEs of SCC-SCs show high overlap with those of ETS2 (T72D) Epi-SC. (D) Heatmap showing H3K27ac ChIP-seq read densities in the SCC-SC SEs. Note that read densities of ETS2 (T72D) induced Epi-SCs are higher than those of control Epi-SCs. (E) Examples of SEs acquired in ETS2 (T72D) induced Epi-SC and which show significant overlap with SCC-SC SEs. Shown are *Elk3* and *Mapk6* loci. (F) Examples of SEs shared not only by SCC-SCs and ETS2 (T72D)-EpiSCs, but also by wild-type EpiSCs. Note that both *Neat1* and *Cdh1* are highly expressed in both normal and malignant skin epithelia. (G) qPCR fold enrichment of ETS2 and myc ChIP DNA of SCC-SC super-enhancer epicenters. Values are normalized to IgG control (n = 3 ± SEM *p<0.05). SC, stem cell; SCC, squamous cell carcinoma.

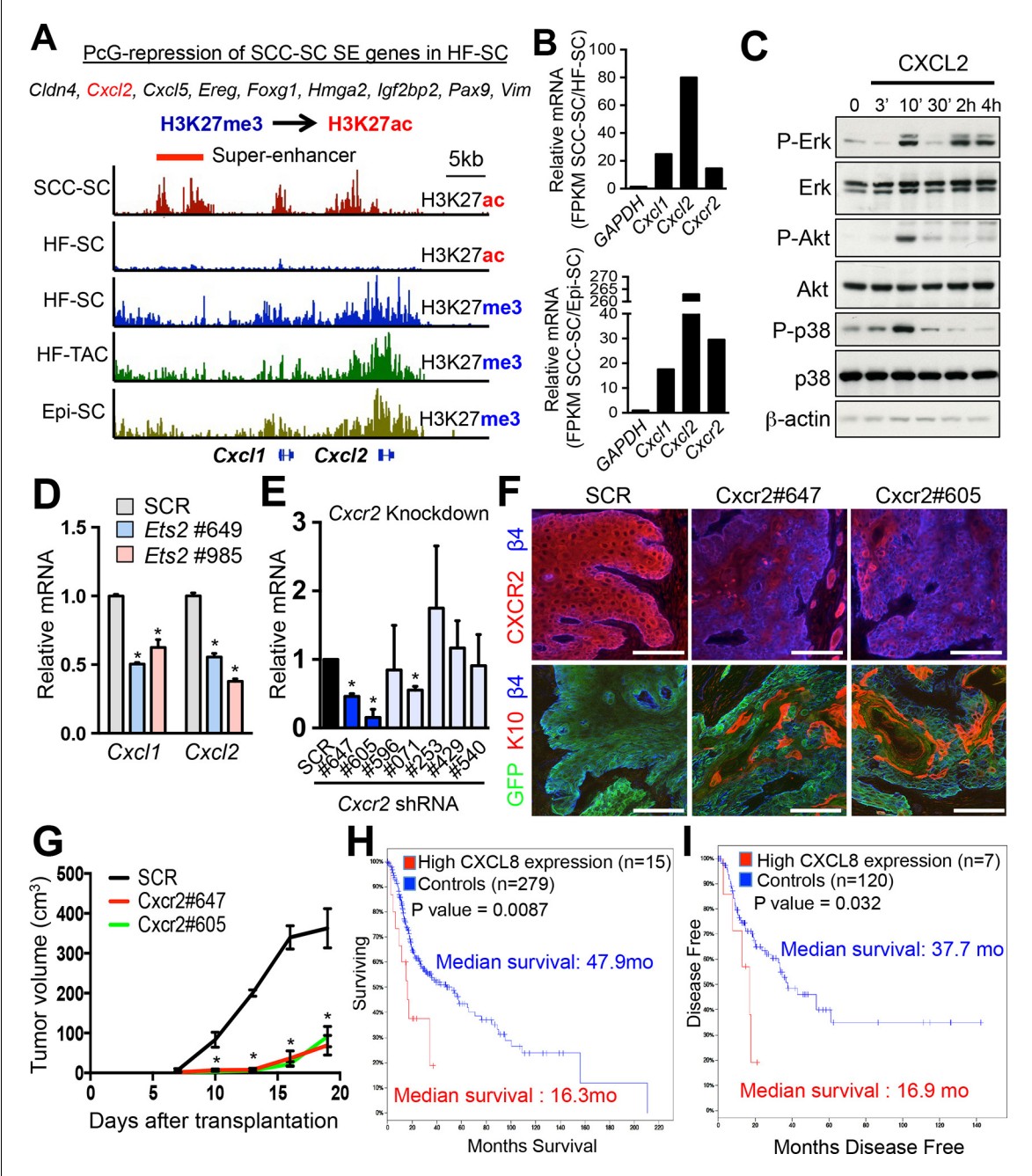

**Figure 7.** Inflammatory mediators are SCC-SE-regulated and affect SCC growth. (**A**) Chromatin status at the *Cxcl1/2* locus in SCC-SCs versus normal skin progenitors. Note that strong peaks of H3K27ac-associated, active chromatin are present throughout this locus in SCC-SCs, while in normal skin progenitors, the locus is heavily marked by H3K27me3, indicative of Polycomb-mediated repression. (**B**) Gene expression changes of *Cxcl1, Cxcl2,* and *Cxcr2* (encoding the receptor for CXCL1/2) in SCC-SCs compared to their normal HF-SC or Epi-SC counterparts. (**C**) Immunoblot analysis shows that CXCL2 activates MAPK signaling in *HRas*$^{G12V}$; *Tgfbr2*-null keratinocytes. β-actin levels are shown as controls. (**D**) Knock-down of *Ets2* reduces *Cxcl1/2* mRNA expression. (**E**) Knock-down efficiency of *Cxcr2* shRNAs in *HRas*$^{G12V}$; *Tgbr2*-null cells as measured by quantitative RT-PCR. (n = 3 ± SEM *p<0.05). (**F**) Immunofluorescence images of allograft tumors (GFP) from *Cxcr2*-shRNA and scrambled control-shRNA transduced SCC-SC cells. Note that CXCR2-reduction is accompanied by a marked increase in K10+ differentiated cells within the tumors. Scale bars, 100 μm. (**G**) Changes in tumor volume of *Cxcr2* knock-down and control allografts over time (n = 3). (**H** and **I**) High *CXCL8* expression (the closest human homologue for CXCL1/2) correlates with shortened survival in human head & neck SCC patients. Kaplan–Meier analysis comparing overall survival (**H**) and disease-free survival (**I**) of TCGA HNSCC patients stratified according to the highest (>5th percentile) *CXCL8* expression/amplification versus the rest (>5th percentile) (please visit http://bit.ly/1Afq0Gt). SC, stem cell; SCC, squamous call carcinoma.

*Figure 7 continued on next page*

*Figure 7 continued*

The following figure supplement is available for figure 7:

**Figure supplement 1.** SCC-SC TFs coordinately bind to *Cxcl1/2* SE-epicenters.

*and F* and *Figure 7—figure supplements 1C and D*, knockdowns were effective in abrogating both mRNA and CXCR2 protein in vitro as well as CXCR2 protein expression in vivo. Moreover, when tumor growth was monitored, a marked reduction of tumor growth and increased differentiation of cancer cells was observed in the allografted tumors (*Figures 7F and G*). These findings were in good agreement with studies on *Cxcr2* knockout mice (*Cataisson et al., 2009*), and confirmed the importance of keratinocyte-specific CXCR2 and autocrine CXCL1/2 signaling in SCCs.

The closest human ortholog of the murine *Cxcl*1 and *Cxcl2* genes is human *CXCL8*, encoding interleukin 8 (IL-8) (*Zlotnik et al., 2006*). Upon analysis of the TCGA database, IL8 expression correlated inversely with the overall survival and disease-free state of head and neck SCC patients (*Figure 7H and I*). Taken together, our data provided compelling evidence that the CXCL1/2-CXCR2 signaling pathway contributes to SCC maintenance and that during tumor progression, it becomes activated through dramatic remodeling of the *Cxcl1/2* locus from a PcG-repressed to an ETS-regulated, SE-activated state.

## Discussion

SCCs are the most common cancers world-wide, occurring most frequently on the skin and affecting millions of people each year. Being visible, skin SCCs are typically treated early and mortality is low. However, SCCs can also arise in the lung, breast, esophagus, cervix, and head and neck, where they are associated with high risk of metastasis and drug resistance. Understanding the molecular complexities that underlie the tumor-initiating SCs within SCC cancers is therefore of paramount importance in developing new and improved diagnostic and therapeutic tools to establish new treatments.

Like HF-SCs and EpiSCs from which SCC-SCs can derive (*Lapouge et al., 2011*; *White et al., 2011*), SCC-SCs are integrin-rich and reside at the interface with their surrounding stroma (*Schober and Fuchs, 2011*; *Lapouge et al., 2012*). However, the gene expression program of these malignant SCs is markedly divergent from their normal skin counterparts (*Schober and Fuchs, 2011*; *Lapouge et al., 2012*). Despite this knowledge, little was known about the master transcriptional regulatory network and the chromatin dynamics that underlie these dramatically different programs of gene expression in SCC-SCs versus normal SCs.

We profiled the SEs of SCC-SCs and compared them to their wild-type counterparts in order to gain a better understanding of the epigenetic rewiring that might be associated with SCC transcription. Since the chromatin landscape in general, and SEs in particular, are highly sensitive to their microenvironment (*Adam et al., 2015*; *Lavin et al., 2014*), it was essential to carry out our experiments on SCC-SCs purified directly from the tumor, that is, in their native niche at the tumor-stroma interface.

In previous studies on SCs in which chromatin landscapes regulated by master regulators have been mapped, ChIP-seq analysis of the known master TFs has preceded subsequent SE analysis. Thus, with cultured ESCs, ChIP-seq for the established pluripotency regulators SOX2, OCT4, NANOG, and KLF4, was already available when ensuing SE analyses revealed that the pluripotency factors bind within these open chromatin domains (*Whyte et al., 2013*; *Dowen et al., 2014*). Similarly for HF-SCs that were purified directly from skin, in vivoChIP-seq analyses had been performed on NFATc1, TCF3/4, SOX9, and LHX2, before subsequent H3K27ac ChIP-seq analysis revealed that these master regulators bind within the HF-SC-specific epicenters of these large open chromatin domains (*Adam et al., 2015*).

For SCC-SCs, global ChIP-seq chromatin mapping in vivohas not been performed, due to the relative paucity of tumor tissue relative to normal skin. Moreover, the abundance of keratin and resilience of the epidermal plasma membrane poses technical hurdles for newer ATAC-seq methods which can be performed with fewer cell numbers and allow TF mapping for some cell types (*Buenrostro et al., 2015*). Compounding the lack of knowledge on chromatin landscapes of SCC-

SCs is the paucity of functional studies on putative transcriptional regulators of these cells, thus far limited to SOX2 (*Siegle et al., 2014*; *Boumahdi et al., 2014*). Thus, in addition to exploring the chromatin landscape of SCC-SCs, a major quest in the present study was to see whether through in vivoH3K27ac-mediated epigenetic analyses of purified SCC-SCs, we might be able to gain new insights into key transcriptional regulators that govern SCC-SC behavior in the context of their native, aberrant SC niche.

To approach the problem, we identified the TFs whose expression is upregulated in SCC-SCs and which display putative TF binding sites within the epicenters of dense H3K27ac peaks of the majority of SCC-SC-specific SEs. As learned from analyses of ESCs and HF-SCs, an additional frequent, albeit not absolute, feature of SC master regulators is that their genes are often regulated by SEs themselves, thereby establishing a feed-forward loop for maintaining the SC state (*Whyte et al., 2013*; *Adam et al., 2015*). By SE profiling of SCC-SCs, identifying putative binding sites within SCC-SC SE epicenters and RNA-seq, we generated a list of candidates for SE-associated master regulator genes.

Several lines of evidence suggest that our strategy was successful. First and foremost was the finding that SOX binding sites were present in >70% of all SCC-SC SEs, a feature predicted from SOX2's functional role in SCCs (*Siegle et al., 2014*; *Boumahdi et al., 2014*). Additionally, given the broad importance of AP-1 (FOS and JUN family) members in cancers (*Eferl and Wagner, 2003*), binding sites for these factors in SCC-SC SEs were notable. Finally, although ETS proteins have not been previously implicated in SCCs, the presence of these sites in nearly 80% of SCC-SC SEs, coupled with the association of *Elk3* and *Ets2* with SEs, begged for functional studies. Indeed, our analyses provide compelling evidence that these factors drive hyperproliferation and SCC progression. Our added finding that high ELK3 and ETS2 expression correlates with poor prognosis in human SCCs fuels the importance of this family of proteins in these cancers.

The role of ETS proteins is particularly intriguing given that their levels matter not only in tumorigenesis but also in maintaining SCC-SC master regulators. Thus when ETS2 was reduced in SCC-SCs, KLF5 and SOX2 were also absent from the resulting cellular masses, and when constitutively active ETS2 was induced in normal epidermis of juvenile mice, the epidermis acquired marked features of hyperproliferation and invasion, accompanied by upregulation/induction of FOS, JUNB and KLF5 and a shift in SE dynamics from one of normal skin progenitors to one of malignant transformation. When coupled with the fact that *Fos*, *Junb* and *Klf5* are associated with SCC-SC SEs which contain AP1, ETS and KLF binding motifs, these findings underscore the physiological relevance of these factors not only in SCC oncogenesis but also in orchestrating the chromatin dynamics of SCC-SCs.

Our results are particularly intriguing in that phosphorylation of ETS2 is known to be regulated by Ras/MAPK signaling (*Foulds et al., 2004*; *O'Neill et al., 1994*; *Brunner et al., 1994*; *Rebay and Rubin, 1995*). Indeed, it was recently shown that 86% of all skin SCCs induced by classical chemical carcinogenesis with 9,10-dimethyl-1,2-benzanthracene (DMBA) involve mutations in either HRas or KRas (*Nassar et al., 2015*), and KRas is markedly upregulated even in SCCs that do not directly involve oncogenic HRas transformation (*Schober and Fuchs, 2011*; *Lapouge et al., 2012*). While cell culture experiments have documented important roles for Ras/MAPK-activated ETS proteins in cellular transformation of NIH3T3 fibroblasts (*Foos et al., 1998*), our findings now lend in vivo relevance to this connection in SCC-SCs.

Given the well-established links between AP1 family members, Ras/MAPK and cancer (*Eferl and Wagner, 2003*), the presence of AP1 and ETS motifs in nearly 80% of SCC-SC SEs takes on all the more significance. Interestingly, many of the ETS binding motifs in SCC-SC epicenters are in close proximity with AP-1 motifs. Closely juxtapositioned ETS-AP1 motifs, essential for cooperative binding, have also been observed in a prostate cancer cell line in culture, and notably, they exist within the regulatory regions of several key oncogenes in which the Ras/MAPK pathway is activated (*Hollenhorst, 2011*). While the large majority of prostate cancers are adenocarcinoma and not SCCs, these parallels are intriguing and merit further investigation in the future.

In addition to gaining insights into chromatin dynamics in SCCs, the SE-regulated gene list that we unearth here is rich in important oncogenes. These include not only *Fos*, *Junb*, *Ets2*, and *Elk3*, but also *Myc*, *Src*, *Mapk6*, *Map2k2*, *Cd44*, and *Tgfa*. Also present on this list are HMGA2, RUNX1 and FOXG1, which are critical TFs for self-renewal of SCC-SCs and other cancer models (*Sgarra et al., 2004*) (*Manoranjan et al., 2013*; *Scheitz et al., 2012*), as well as KLF5, which plays a

role in epithelial cell hyper-proliferation under inflammatory conditions (*Sur, 2006*). Moreover, HIF1α is a key regulator of hypoxic response in cancer (*Wilson and Hay, 2011*). Overall, the list of SCC-SC SE-associated genes was rich in important cell signaling genes that promote hyper-proliferation, migration, invasion, inflammation and cancer metabolism. When taken together with our findings that the SEs regulating normal HF-SC master regulators are decommissioned in SCC-SCs, these data expose SEs as chromatin gatekeepers of the cancer SC state.

In HF-SCs, the SE chromatin landscape is highly sensitive to the SC niche microenvironment (*Adam et al., 2015*). In this regard, the preponderance of SCC-SC SEs associated with inflammatory genes was intriguing since inflammation in the perivascular stroma is a key feature of the SCC microenvironment (*Oshimori et al., 2015*). Inflammatory players can be proangiogenic, growth-promoting or tumor suppressive. In addition, epithelial-derived factors collectively termed as the 'epimmunome' (*Swamy et al., 2010*) can instruct immune cells, thereby harboring the potential to impinge upon immunocompetence and tumor immunosurveillance. Our discovery that SCC cells express and secrete CXCL1 and CXCL2, appears to be particularly relevant, because this phenomenon turned out to be hardwired through the establishment of SEs. Additionally, CXCL1/2 can signal to the immune system, solidifying their part in the epimmunome. Intriguingly, however, we found that SCC cells also express CXCR2, the cognate receptor for CXCL1/2, thus establishing an autocrine loop to fuel epithelial proliferation. Indeed, when we knocked down *Cxcr2,* tumor growth was severely impaired.

In summary, we have unearthed many new insights regarding SEs that go beyond mere cancer-specific gene expression. For the first time, we show that in vivo SEs of a cancer SC are markedly distinct from their normal counterparts, and they reflect their dramatically altered microenvironment. We show that SEs controlling the normal SC-TFs are decommissioned in cancer, and that the new SEs drive cooperative auto-regulation of novel master regulators that specify the cancer state. For SCC, this includes an ETS2/ELK3-AP1-SOX2-KLF5 network of which level of ETS family members appears to be critical in regulating SCC SE dynamics and orchestrating expression of a cohort of oncogenic, growth-promoting, and epimmune genes. Of additional relevance are the mutually exclusive H3K27 modifications of SEs and PcG-silencing that provide a powerful two-way switch for the cancer-normal SC balance.

## Materials and methods

### Mouse lines

Female CD1 mice (Charles River, New York, NY) were used for the purification of HF SCs. Female CD1 mice transgenic for krt14-H2B-GFP (*Tumbar, 2004*) were used for the purification of TACs. We used *Tgfbr2* floxed (*Leveen, 2002*) mice to isolate primary keratinocytes. Nude mice were from Charles River Laboratories. For lentiviral injections, transduced mice were confirmed by genotyping with RFP primers: forward 5' –ATCCTGTCCCCTCAGTTCCAGTAC-3', reverse 5'-TCCACGATGGT G-TAGTCCTCGTTG-3'. For TRE-mycETS2 or mycETS2 (T72D) transduced mice, positive mice were fed with doxycycline-containing chow, starting at P0. Mice were maintained in the Association for Assessment and Accreditation of Laboratory Animal Care-accredited animal facility of The Rockefeller University (RU), and procedures were performed with Institutional Animal Care and Use Committee (IACUC)-approved protocols (#13622-H, #14693-H and #14765-H).

### Cell culture and generation of transformed cell lines

Newborn, primary mouse epidermal keratinocytes from *Tgfbr2* floxed were cultured on 3T3-S2 feeder layer in 0.05 mM Ca++ E-media supplemented with 15% serum (*Blanpain et al., 2004*). For adenoviral infections with Ad-Cre-GFP and Ad-GFP (1010 pfu/ml; University of Iowa, Gene Transfer Vector Core Iowa), cells were plated in 6-well dishes at 200,000 cells/well and incubated with adenovirus at a MOI of 100 in the presence of polybrene (100 mg/ml) overnight. After 2 days, infected cells were sorted to purity by FACS and expanded in culture for an additional 5 days. Of note, after 5 days cells lost their GFP expression from the transient adenoviral infection. For the following retroviral infections with MSCV-HRasV12-IRES-GFP, cells were again plated in six-well dishes at 200,000 cells/well and incubated with retrovirus at a MOI of about 100 in the presence of polybrene (100 mg/ml) overnight. After 2 days, GFP positive cells were again sorted to purity by FACS and

tested for loss of Tgfbr2 and presence of RasV12 by Western blot and RT-PCR. To generate myg-tagged ETS2 expressing SCC cells, their tumor SCsstem cells were transduced with lentivirus containing CMV-mycETS2-PGK-H2BRFP, and after 3 days, RFP expressing cells were purified by FACS, and then injected into mice to form in vivo SCCs.

## Analysis of human HNSCC patient data

We analysed the publicly available data sets of the The Cancer Genome Atlas (TCGA: http://cancer-genome.nih.gov). The cBioPortal for Cancer Genomics developed and maintained by the Computational Biology Center at Memorial Sloan-Kettering Cancer Center was used to mine the publicly available TCGA dataset on HNSCC (*Gao et al., 2013*; *Cerami et al., 2012*). To retrace the exact Kaplan–Meyer analysis, please visit SurvExpress http://bioinformatica.mty.itesm.mx:8080/Biomatec/SurvivaX.jsp for the analysis of HNSCC patients stratified by the 340 SE-associated genes (*Aguirre-Gamboa et al., 2013*).

## Tumor formation

For allograft transplantation, $1.0 \times 10^5$ mouse primary tumor cells were intra-dermally injected with Matrigel (BD, East Rutherford, NJ) in Nude mice. Tumor size was measured every three days and calculated using the formula $4/3\pi \times L/2 \times W/2 \times D/2$.

## Flow cytometry

Tumors were dissected from mice, and normal skin, blood vessels and connective tissue were removed. Tumor tissues were minced and treated with 0.25% collagenase (Sigma, St. Louis, MO) in HBSS (Gibco, Carlsbad, CA) for 20 min at 37°C with shaking. After washing with cold PBS, the tissues were treated with 0.25% trypsin for 10 min at 37°C, and then were washed with PBS containing 5% of fetal bovine serum (FBS). The cell suspensions were filtered through 70-μm and 45-μm strainers. Isolation of HF-SCs and Epi-SCs from adult mice back skins was done as previously described (*Adam et al., 2015*), Briefly, for telogen skin, subcutaneous fat was removed with a scalpel, and skins were placed dermis side down on trypsin (Gibco) at 37°C for 35 min. Single cell suspensions were obtained by scraping the skin gently. After washing with PBS containing 5% of FBS, cells were filtered through 70-μm and 45-μm strainers. Cell suspensions were incubated with the appropriate antibodies for 10 min on ice. The following antibodies were used for FACS: Integrin α6-PE (1:100, BD Biosciences), Integrin β1-APC/Cy7 (1:200, Biolegend, San Diego, CA), CD34-eFluoro660 (1:100, eBiosciences, San Diego, CA) and Sca-1-PerCP-Cy5.5 (1:100, eBiosciences) and CD11b (1:200, eBiosciences). DAPI was used to exclude dead cells. Cell isolations were performed on FACSAriaII sorters running FACSDiva software (BD Biosciences).

## ChIP-seq

Immunoprecipitations were performed on FACS-sorted populations from tumors and nano-ChIP-seq was done as described(*Adli and Bernstein, 2011*). For each ChIP-seq run, $2 \times 10^7$ cells were used and for nano-ChIP-seq, $1{\sim}2 \times 10^5$ cells were used. Cells were sorted by FACS. Sorted cells were cross-linked in fresh 1% (wt/vol) formaldehyde solution for 10 min at room temperature followed by adding one-tenth volume of 2.5 M Glycine to quench formaldehyde. Cells were rinsed twice with PBS, flash frozen in liquid nitrogen and stored at -80°C prior to use. Cells were resuspended and lysed in lysis buffers. To solublilize and shear cross-linked DNAs, lysates were subjected to a Bioruptor Sonicator (UCD-200, Diagenode, Denville, NJ) according to a 30X regimen of 30 s sonication followed by 60 s rest. The resulting whole-cell extract was incubated overnight at 4°C with Dynabeads Protein G magnetic beads (Life Technologies, Carlsbad, CA) which had been pre-incubated with anti-H3K27ac (ab4729, abcam, Cambridge, MA) antibody. After ChIP, samples were washed with low salt, high salt, LiCl and Tris-EDTA buffer for 10 min. Bound complexes were eluted and crosslinking was reversed by overnight incubation at 65°C. Whole cell extract DAN was also treated for cross-link reversal.

ChIP DNA was prepared for sequencing by repairing sheared DNA and adding Adaptor Oligo Mix (Illumina, San Diego, CA) in the ligation step. A subsequent PCR step with 25 amplification cycles added the additional Solexa linker sequence to the fragments to prepare them for annealing to the Genome analyzer flow cell. After amplification, a narrow range of fragment sizes between

150–300 bp was selected by separation on a 2% agarose gel and the DNA was gel-purified and diluted to 10 nM for loading on the flow cell. Nano-ChIP-seq was carried out similarly, except that ChIP DNA was primed with Sequenase enzyme using the primer 1 that contains universal sequence, the BciVI restriction site and random 9-mer and then amplified using primer 2 that contains universal sequence (Primer 1: 5′-GACAT<u>GTATCC</u>GGATGTNNNNNNNNN-3′, Primer 2: 5′-GACAT<u>GTATCC</u>G-GATGT-3′). After BciVI digestion, the ChIP DNA was ligated with Illumina adapters and amplified, and fragment sizes between 200-700bp were selected by electrophoresis. The DNA was gel-purified and sequencing was performed on the Illumina HiSEq 2500 Sequencer following manufacturer protocols. ChIP-seq reads were aligned to them mouse genome (mm9, build 37) using Bowtie aligner (*Langmead et al., 2009*). ChIP-seq signal tracks were presented by Integrative Genomics Viewer (IGV) software.

## SE profiling

H3K27ac peaks were called by the program MACS (*Zhang et al., 2008*) (v1.4.2, default parameters) from the aligned ChIP-seq data with the input as controls. The peaks were associated to genes in the mouse RefSeq annotation; those located within 2 kb of transcription start site were called as 'promoter' peaks and the rest collectively considered as 'enhancer' peaks. The H3K27ac enhancer peaks from two biological replicates were merged and were used for the identification of SEs, using the algorithm described previously, wherein enhancer peaks were stitched together if they are located within 12.5 kb of each other and if they don't have multiple active promoters in between. The stitched enhancers were then ranked according to increasing H3K27ac signal intensity (*Whyte et al., 2013*). Enhancer-gene assignments were performed using the following criteria to make gene assignments: (*Whyte et al., 2013*) proximity of genes to the SE of cancer SCs; (*Parker et al., 2013*) high transcriptional activity in cancer SCs by RNA-seq; (*Hnisz et al., 2013*) correlation between SE and candidate gene expression in CSCs and HFSCs. GO function enrichment analyses were conducted by the software GREAT (*McLean et al., 2010*) using the list of SE coordinates and the default setting. For motif analysis of SEs, we searched the 1-kb sequences under the summits of H3K27ac peaks within SEs searched for enriched motifs using the software HOMER (v4.6) with the default setting and genome as background (*Heinz et al., 2010*).

## ChIP-qPCR

Chromatin immunoprecipitations were performed as described above with FACS-sorted populations from tumors, and anti-ETS2 antibody (Santa Cruz, Dallas, TX), anti-myc antibody (Abcam) and normal rabbit IgG (Cell Signaling, Danvers, MA) were used. The purified ChIP DNAs were mixed with indicated primers and SYBR green PCR Master Mix (Sigma), and qRT-PCR was performed on an Applied Biosystems 7900HT Fast Real-Time PCR system. The following primer sequences were used (5′ to 3′):

*Cxcl1/2* SE_epicenter forward: CAACATGCCTAGCCCGTGAGTC,
*Cxcl1/2* SE_epicenter reverse: GTGCCCTGTTTCACAGATAGAGGC
*Elk3* SE_epicenter forward: CGTCCATTCTCTCCCCTTTTCTAGC,
*Elk3* SE_epicenter reverse: CATGATTGGCAGTGGAGTATCGAGC
*Ets2* SE_epicenter forward: CAATGGCTTGGAGATCCCCGAC,
*Ets2* SE_epicenter reverse: CAGGGTCACCAGTGAGTCACAG
*Fam49b* SE_epicenter forward: CCACGGAACCTGAGAATGAAGCC,
*Fam49b* SE_epicenter reverse: GCTTCAACTGACTGAACTCCCAGG
*Il1a/b* SE_epicenter forward:, CAGAGGGTGGCACAGGATAGACAG,
*Il1a/b* SE_epicenter reverse: CAGTGTCCTGCCCAGTCATCTG
*Pthlh* SE_epicenter forward:, CTGAGCTACACCCTTCCACTTCAC,
*Pthlh* SE_epicenter reverse: GTCTTCATTCCTCTGAGCCAATGTGC
Negative control region forward: CATGCAAGCATCACCAACAAAGTA,
Negative control region reverse: CCATGGAACTGGGACCTTCTTC

## RNA purification, RNA-Seq, and qRT-PCR

FACS-isolated cells and cultured cells were directly lysed into TRIzol LS (Invitrogen, Carlsbad, CA). Total RNA was purified using the Direct-zol RNA MiniPrep kit (Zymo Research, Irvine, CA) per

manufacturer's instructions. For RNA-Seq, quality of the RNA for sequencing was determined using an Agilent 2100 Bioanalyzer; all samples used had RNA integrity numbers >9. Library preparation using the Illumina TrueSeq mRNA sample preparation kit was performed at the Weill Cornell Medical College Genomic Core facility (New York, NY), and RNAs were single-end sequenced on Illumina HiSeq 2000 machines. Alignment of reads was done using Tophat with the mm9 build of the mouse genome. Transcript assembly and differential expression was determined using Cufflinks with Refseq mRNAs to guide assembly (*Trapnell et al., 2010*). Analysis of RNA-seq data was done using the cummeRbund package in R (*Trapnell et al., 2012*). For real-time qRT-PCR, equivalent amounts of RNA were reverse-transcribed by SuperScript VILO cDNA Synthesis Kit (Life Technologies). cDNAs were normalized to equal amounts using primers against Ppib2, Hprt, or Gapdh. cDNAs were mixed with indicated primers and SYBR green PCR Master Mix (Sigma), and qRT-PCR was performed on an Applied Biosystems 7900HT Fast Real-Time PCR system. The following primer sequences were used (5' to 3'):

*mPpib2* forward: GTGAGCGCTTCCCAGATGAGA,
*mPpib2* reverse: TGCCGGAGTCGACAATGATG
*mHprt* forward: GATCAGTCAACGGGGGACATAAA,
*mHprt* reverse: CTTGCGCTCATCTTAGGCTTTGT
*mGapdh* forward: GTCGTGGAGTCTACTGGTGTCTTCAC,
*mGapdh* reverse: GTTGTCATATTTCTCGTGGTTCACACCC
*mEts2* forward: GACGGGGGATGGATGGGAGTTCAAG,
*mEts2* reverse: AGCCCAGCAAGTTCTGCAGGTCACA
*mElk3* forward: TCCTCACGCGGTAGAGATCAG,
*mElk3* reverse: GTGGAGGTACTCGTTGCGG
*mCxcr2* forward: TGTTCTGCTACGGGTTCACACTG,
*mCxcr2* reverse: GCGGCGCTCACAGGTCTC
*mCxcl1* forward: CCACACTCAAGAATGGTCGC,
*mCxcl1* reverse: TCTCCGTTACTTGGGGACAC
*mCxcl2* forward: CGGTCAAAAAGTTTGCCTTG,
*mCxcl2* reverse: TCCAGGTCAGTTAGCCTTGC

## Histology, immunofluorescence, and imaging

For immunofluorescence microscopy of tumor sections, dissected tumors were fixed with 1% PFA in PBS for 1 hr at 4°C, washed with PBS, incubated with 30% sucrose overnight at 4°C, and embedded in OCT (VWR, Radnor, PA). For telogen, anagen and phosphomimic Ets2 (T72D) overexpressed skin, tissues were harvested and embedded in OCT. Cryosections were cut at a thickness of 10 μm on a Leica cryostat and mounted on SuperFrost Plus slides (VWR). Sections were blocked for 1 hr in blocking buffer (5% normal donkey serum, 1% BSA, 2% fish gelatin, 0.3% Triton X-100 in PBS). Primary antibodies were diluted in blocking buffer and incubated at 4°C overnight. The following primary antibodies were used: ETS2 (rabbit, 1:500, Aviva Systems Biology, San Diego, CA; or rabbit, 1:500, Life technologies), Phospho-ETS2 pThr72 (rabbit, 1:500, Life technologies), ELK3 (rabbit, 1:100, Novus, Littleton, CO), KLF5 (goat, 1:50, R&D, Minneapolis, MN), SOX2 (rabbit, 1:200, Abcam), Phospho-NFkB p65 pSer276 (rabbit, 1:100, Cell Signaling), FOS (rabbit, 1:100, Abcam), Integrin β4 (rat, 1:100, BD Pharmingen), GFP (chicken, 1:2,000, Abcam), pSmad2 (rabbit, 1:1,000, Cell Signaling), CD34 (rat, 1:100, BD Pharmingen), K6 (guinea pig, 1:5,000, Fuchs laboratory), K5 (guinea pig, 1:1,000, Fuchs laboratory), K10 (rabbit, 1:1,000, Covance, Princeton, NJ), Laminin5 (rabbit, 1:1,000, Fuchs Laboratory), CXCR2 (rabbit, 1:200, Santa Cruz), Loricrin (rabbit, 1:1000, Fuchs Laboratory), CD31 (hamster, 1:200, Millipore, Billerica, MA), and Ki67 (rabbit, 1:1000, Novocastra, Buffalo Grove, IL). After washing with PBS, sections were treated for 1 hr at room temperature with secondary antibodies conjugated with Alexa 488, RRX, or Alexa 647 (Life Technologies). Slides were washed, counterstained with 4'6'-diamidino-2-phenilindole (DAPI), and mounted in Prolong Gold (Life Technologies). Images were acquired with an Axio Observer.Z1 epifluorescence microscope equipped with a Hamamatsu ORCA-ER camera (Hamamatsu Photonics, Geldern, Germany), and with an ApoTome.2 (Carl Zeiss, Oberkochen, Germany) slider that reduces the light scatter in the fluorescent samples, using 20X objective, controlled by Zen software (Carl Zeiss). Z stacks were projected and RGB images were assembled using ImageJ. Panels were labelled in Adobe Illustrator CS5.

## Immunohistochemistry and histological analyses of human tumors

Immunohistochemistry was performed as previously described (Bian et al., 2009). Briefly, 5-μm sections were cut, stained with H&E or processed for immunohistochemistry/immunofluorescence microscopy. For immunoperoxidase staining, paraffin-embedded sections were dehydrated and antigenic epitopes exposed using a 10-mM citrate buffer (pH 6.0) in a pressure cooker. Sections were incubated with the following primary antibodies at 4°C overnight: rabbit anti-ETS2 (1:100; LifeTechnologies PA5-28053) and rabbit anti-phospho-ETS2 (1:100; LifeTechnologies 44-1105G). Primary antibody staining was visualized using peroxidase-conjugated anti-rabbit IgG followed by the DAB substrate kit for peroxidase visualization of secondary antibodies (Vector Laboratories, Burlingame, CA). Tissue microarrays comprising healthy human skin samples, human skin SCCs as well as head and neck SCCs (HNSCC) were obtained from US Biomax, Rockeville, MD: SK241, SK801b, SK811a, SK2081, and HN803a.

## Lentiviral expression constructs

To generate the phosphomimic Ets2 (T72D) expression construct, Ets2 cDNA was PCR amplified and 72 threonine residue was mutated to glutamate by site-directed mutagenesis. Final PCR product was nserted into the LV-TRE-PGK-H2BmRFP1 construct. The resulting LV-TRE-mycEts2(T72D)-PGK-H2BmRFP was used for in utero injections.

## Lentivirus production and transduction

Production of VSV-G pseudotyped lentivirus was performed by calcium phosphate transfection of 293FT cells (Invitrogen) with pLKO.1 and helper plasmids pMD2.G and psPAX2 (Addgene plasmid 12259 and 12260, Cambridge, MA). Viral supernatant was collected 46 hr after transfection and filtered through a 0.45-μm filter. For lentiviral infections in culture, cells were plated in 6-well plate at $1.0 \times 10^5$ cells per well and incubated with viruses in the presence of polybrene (20 μg/ml) for 30 min, and then plates were spun at 1100 g for 30 min at 37°C in a Thermo IEC CL40R centrifuge. Infected cells were selected with puromycin. For in vivo lentiviral transduction, viral supernatant was filtered (0.45-μm filter) and concentrated by ultracentrifugation. Final viral particle was resuspended in viral resuspension buffer (20 mM Tris pH 8.0, 250 mM NaCl, 10 mM $MgCl_2$, 5% sorbitol) and 1 or 0.5 μl was in utero injected into E9.5 embryos.

## shRNA sequences for gene knockdown

For knockdown experiments, we used clones from the Broad Institute's Mission TRC mouse library. We tested the knockdown efficiency of 5–10 independent shRNAs for each gene and used the following clones and target sequence:

The scramble shRNA (Sigma SHC002, CAACAAGATGAAGAGCACCAA), *Ets2* (TRCN0000233985, CATTGATAAAGAGCCGTTATA; TRCN0000042649, CCGTCAATGTCAATTACTGTT), *Elk3* (TRCN0000042643, GCTGAGATACTATTACGACAA; TRCN0000235780, ATCAGGTTTGTGACCAATAAA; TRCN0000235783, AGAGCGCTGAGATACTATTAC), *Cxcr2* (TRCN0000026605, GCCTTGAATGCTACGGAGATT; TRCN0000026647, CGTTACAATTACAGTGAGATA).

## Statistics

Data were analyzed and statistics performed using unpaired two-tailed Student's t-test in Prism6 (GraphPad software, La Jolla, CA). Significant differences between two groups were noted by asterisks or actual p values. Quantification data were presented in mean value ± SEM or in box and whisker plots with the dimensions of the box encompassing the 25th–75th percentile, the horizontal bar representing the median, and the error bars representing minimum and maximum values.

ChIPseq and RNAseq databases were deposited in GEO (Accession #GSE72147).

## Acknowledgements

We thank members of the Fuchs' laboratory for helpful discussions. We thank S Karlsson for TβRII[fl/fl] mice and James A Fagin for HRas[LSL-G12V] mice. HY is a recipient of a Kwanjeong Educational Foundation graduate fellowship. RA is a recipient of an Anderson Cancer Center graduate fellowship. DS was an Emerald Foundation Young Investigator and was supported by the Human Frontier Science

Program Organization (HFSP LT000793/2012). EF is an Investigator of the Howard Hughes Medical Institute. This work was supported by grants to EF from the National Institutes of Health (R01-AR31737) and NYSTEM #CO29559.

## Additional information

### Competing interests
EF: Reviewing editor, *eLife*. The other authors declare that no competing interests exist.

### Funding

| Funder | Grant reference number | Author |
|---|---|---|
| National Institutes of Health | R01-AR-31737 | Elaine Fuchs |
| Howard Hughes Medical Institute | | Elaine Fuchs |
| Human Frontier Science Program | LT000793/2012 | Daniel Schramek |
| Anderson Cancer Center | Graduate Student Fellowship | Rene C Adam |
| Kwanjeong Educational Foundation | Graduate Student Fellowship | Hanseul Yang |
| New York Stem Cell Foundation | #CO29559 | Elaine Fuchs |

The funders had no role in study design, data collection and interpretation, or the decision to submit the work for publication.

### Author contributions
HY, DS, conceived and designed the project, contributed to the experiments and acquisition of data and the interpretation, wrote the manuscript and participated in reviewing and approving the final version of the manuscript., Conception and design, Acquisition of data, Analysis and interpretation of data, Drafting or revising the article; RCA, contributed to the experiments and acquisition of data and participated in reviewing and approving the final version of the manuscript., Acquisition of data, Analysis and interpretation of data, Drafting or revising the article; BEK, PW, contributed to the data analyses and participated in reviewing and approving the final version of the manuscript., Analysis and interpretation of data, Drafting or revising the article; DZ, contributed to the interpretation and the data analyses, wrote the manuscript and participated in reviewing and approving the final version of the manuscript., Analysis and interpretation of data, Drafting or revising the article; EF, conceived and designed the project, contributed to the interpretation, wrote the manuscript and participated in reviewing and approving the final version of the manuscript, Conception and design, Analysis and interpretation of data, Drafting or revising the article

### Ethics
Human subjects: Tissue microarrays comprising healthy human skin samples, human skin SCCs as well as head and neck SCCs (HNSCC) were obtained from US Biomax, Rockeville. All tissue is collected under the highest ethical standards with the donor being informed completely and with their consent. The company followed standard medical care and protect the donors' privacy. All human tissues are collected under HIPPA approved protocols.
Animal experimentation: Mice were maintained in the Association for Assessment and Accreditation of Laboratory Animal Care-accredited animal facility of The Rockefeller University (RU), and procedures were performed with Institutional Animal Care and Use Committee (IACUC)-approved protocols (#13622-H, #14693-H and #14765-H).

# Additional files

## Supplementary files

• Supplementary file 1. List of super-enhancers of SCC-SC in vivoby H3K27ac ChIP-seq: chromosomal coordinates and corresponding gene assignments.

• Supplementary file 2. List of super-enhancers of SCC-SC in vivo by H3K27ac nano-ChIP-seq: Chromosomal coordinates and corresponding gene assignments.

• Supplementary file 3. List of super-enhancers of ETS2(T72D) super-activated Epi-SC in vivo by H3K27ac nano-ChIP-seq: Chromosomal coordinates and corresponding gene assignments.

## Major datasets

The following datasets were generated:

| Author(s) | Year | Dataset title | Dataset URL | Database, license, and accessibility information |
|---|---|---|---|---|
| Yang H, Fuchs E | 2015 | ETS Family Transcriptional Regulators Drive Chromatin Dynamics and Malignancy in Squamous Cell Carcinomas | http://www.ncbi.nlm.nih.gov/geo/query/acc.cgi?acc=GSE72147 | Publicly available at the NCBI Gene Expression Omnibus (Accession no: GSE72147). |

The following previously published dataset was used:

| Author(s) | Year | Dataset title | Dataset URL | Database, license, and accessibility information |
|---|---|---|---|---|
| Adam RC, Fuchs E | 2015 | Pioneer factors govern super-enhancer dynamics in stem cell plasticity and lineage choice | http://www.ncbi.nlm.nih.gov/geo/query/acc.cgi?acc=GSE61316 | Publicly available at the NCBI Gene Expression Omnibus (Accession no: GSE61316). |

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
