## [Decision Letter]

Thank you for submitting your study titled, "ETS family of transcriptional regulators drive chromatin dynamic and malignancies in squamous cell carcinomas" for consideration by *eLife*. Your manuscript has been reviewed by two experts in the field and by a member of the Board of Reviewing Editors (BRE). We have appended the detailed reviews for you at the bottom of this letter.

As you can see, the reviewers and the BRE member find the study of interest to the journal and are in agreement that the initial profiling of HF-SC's and SCC-SC's for histone modifications are interesting. However, the reviewers unanimously agree that the lack of genome-wide data testing whether the identified transcription factors are actually acting by regulating super enhancers under growth and malignancy conditions has lessened their enthusiasm for the manuscript. We would be interested in a revised study if you have performed these control experiments and have the genome-wide data and the analyses available so that they can be included in a revised manuscript. In this case we, would be happy to consider such a revised manuscript. Otherwise, we feel that the requested studies will take much longer than allowed for a revised submission at *eLife*.

Reviewer #1:

Recently super enhancers have been revealed as critical cis-regulatory elements to maintain cell fate as well as to determine cell lineages. A role of super enhancers in cancer cells has not been well characterized. In this work, Yang and colleagues using squamous cell carcinoma (SCC) to interrogate a role of super enhancers in cancer cells. Overall this is an interesting paper with identification of ETS transcript factor family in SCC, which is subject to the regulation by super enhancers. Furthermore, the ETS TF associated super enhancer is highly correlated with development of SCC and is more specific for SCC but not to their "normal" counterpart HF-SCs. This ETS binding super enhancer also induced a positive forward loop of CXCL1/2 with their corresponding receptors, reflecting a feature of tumor microenvironment. Their method of prospecting for genes involved in various disease states using SE's and binding motifs could prove very useful. All these are novel findings. However, their conclusions on the relevance of SE's still require direct evidence.

Major concern:

Given the fact that super enhancers are so implied throughout the paper, and indeed even in the title, I find it odd that the chromatin landscape isn't examined even once after any of their treatments. Aside from the initial profiling of HF-SC's and SCC-SC's, neither histone modifications nor occupancy of TFs reported as SE regulators are ever explored. This could have been done after ELK3, ETS2, or CXCR2 knockdown, but instead they rely very heavily on the very tenuous assumption that because they have binding sites, and their genes are regulated by SE's, that any effects of their treatments must directly involve SE's. Given the pleiotropic effects any of these targets could have, relying solely on phenotypic outcome to imply a specific mechanism of action seems odd.

Reviewer #2:

In this manuscript, the authors did a comparison study on super-enhancers in squamous cell carcinoma stem cells and hair follicles stem cells. Based on the bioinformatics analysis on the H3K27ac ChIP-seq data and RNA-seq data, they identified both known and new cancer stem cell specific transcription factors under the super-enhancers. They functionally studied ETS family factors by both knocking down and ORF rescue experiments, and the ETS family factors played significant roles in driving squamous cell carcinoma stem cells. However, there is a lack of solid biological evidence that the ETS factors are actually regulating super-enhancers and that feedback on super-enhancers regulates other factors like *Cxcl1* and *Cxcl2*. Currently, the relationship between supper-enhancer dynamics and ETS family proteins is uncertain.

Major concerns:

1) The authors performed H3K27ac ChIP-seq analysis on SCC-SC in comparison with the one in HF-SC, and identified cancer stem cell specific SEs and SEs associating genes. To better understand the specificity of super-enhancers in cancer stem cells, the authors should include one more SCC sample control.

2) In subheading “SE-associated ETS TFs constitute cancer vulnerabilities within SCCs”, the authors wrote: "Our ChIP-seq analyses showed that genes encoding both ELK3 and ETS2 were governed by super-enhancers. Given that both of these TFs are known to bind to ETS sequence motifs, we wondered whether they might both participate in governing super-enhancer dynamics." But, the follow up experiments are all on the functional study of *Elk3* and *Ets2* in tumorigenesis by knocking down and cDNA rescue, which cannot address the initial question. A correct experiment is *Elk3* ChIP-seq and *Ets3* ChIP-seq. Since the authors also appealed that they cannot perform ChIP-seq on these two factors due to the limitation of cancer tissues, they should change this sentence.

3) The subtitle "Testing the physiological significance of super-enhancer dynamics in SCCs" was over-claimed, please change it accordingly to the data.

4) In paragraph two of the subsection “SE-associated ETS TFs constitute cancer vulnerabilities within SCCs”, the authors claimed *Elk3* transcription was knocked down by >90%, while in Figure 3 the efficiency is not that high.

5) Similarly, in the fourth paragraph of the aforementioned section, the authors claimed *Ets2* transcription was knocked down by 80-90%, while, in Figure 4, the efficiency does not appear to be that high either.

6) In "SE-associated ETS TFs constitute cancer vulnerabilities within SCC", the authors started from the purpose to explore whether *Elk3* and *Ets2* participate in governing super-enhancer dynamics, but turned out to test the functional significance of them. I would suggest changing the first two sentences to focus more on their function.

7) The title of the paper concludes that ETS family transcriptional regulators drive chromatin dynamics. However, in the paper, due to the difficulties to profile global chromatin states by ChIP-seq, no direct proof can be found to fully support this conclusion. Without direct evidence to support the major conclusion of the paper it is not of sufficient impact for *eLife*. It should be noted that the existence of a binding motif does not necessarily indicate the binding of a transcription factor. The regulation of the recruitment of transcription factors has been shown by many studies.

---

## [Author Response]

Reviewer #1:

Major concern:

Given the fact that super enhancers are so implied throughout the paper, and indeed even in the title, I find it odd that the chromatin landscape isn't examined even once after any of their treatments. Aside from the initial profiling of HF-SC's and SCC-SC's, neither histone modifications nor occupancy of TFs reported as SE regulators are ever explored. This could have been done after ELK3, ETS2, or CXCR2 knockdown, but instead they rely very heavily on the very tenuous assumption that because they have binding sites, and their genes are regulated by SE's, that any effects of their treatments must directly involve SE's. Given the pleiotropic effects any of these targets could have, relying solely on phenotypic outcome to imply a specific mechanism of action seems odd.

We agree with the reviewer and have addressed histone modification as well as TF occupancy. Most significantly, we carried out in vivo RNA-seq and H3K27ac ChiP-seq experiments on purified skin basal epidermal control progenitors and progenitors following Doxy-induction of skin epidermis to induce a phosphomimetic super-activated form of ETS2, which normally requires oncogenic HRas-MAP kinase for this threonine phosphorylation. Not only does the epidermis transform into an SCC-like state, but at the transcriptional level, many of the genes upregulated in SCC-SCs are now elevated in the ETS skin progenitors. At the level of chromatin landscaping, this not only includes the SE-regulated genes, but in fact these genes now acquire the chromatin remodeling and super-enhancer acquisition that we observed in SCC-SCs. Finally, although the numbers of cells required for ETS ChIP-seq are still prohibitive, as are the numbers of cells for ETS2 knock-down tumors, we were successful in carrying out ETS2 ChiP-qPCR with a number of SE-regulated genes from these cells purified in vivo.

It is remarkable that the phosphomimetic ETS2 induces a phenotype and an SE profile and expression profile that are not only similar to SCC-SEs, but in addition SE-regulated genes, *Cxcl1, Cxcl2* and *Elk3* ranked among the highest upregulated genes, analogous to what we had seen in SCCs (Figure 6). Moreover, 50% of the genes found to be upregulated upon ETS superactivation are also upregulated in SCC-SC (Figure 6). The entire Figure 6 is now new and encompasses these and other exciting data. We thank the reviewer for his/her comments and suggestions.

Reviewer #2:

Major concerns:

*1) The authors performed H3K27ac ChIP-seq analysis on SCC-SC in comparison with the one in HF-SC, and identified cancer stem cell specific SEs and SEs associating genes. To better understand the specificity of super-enhancers in cancer stem cells, the authors should include one more SCC sample control.*

We apologize for not clearly articulating how we did our experiments. We actually *did* perform the CHIP-seq experiments and the analysis in two completely independent experiments. Moreover, every duplicate was a pool of several tumors, which was necessary as we had to FACS-purify 20 million (!) cancer stem cells for a single CHIP experiment. Importantly, the duplicates showed a robust Pearson correlation coefficient of genome-wide read densities of r>0.89 (please see representative example in Figure 1—figure supplement 2). In fact there was striking reproducibility. We think that once the reviewer appreciates this, he/she will agree that a third sample on collected and sorted 20 million cancer stem cells out of ~120 tumors would not only require considerable time and money but will not add additional data/insights. We’ve now clarified the text.

*2) In subheading “SE-associated ETS TFs constitute cancer vulnerabilities within SCCs”, the authors wrote: "Our ChIP-seq analyses showed that genes encoding both ELK3 and ETS2 were governed by super-enhancers. Given that both of these TFs are known to bind to ETS sequence motifs, we wondered whether they might both participate in governing super-enhancer dynamics." But, the follow up experiments are all on the functional study of Elk3 and Ets2 in tumorigenesis by knocking down and cDNA rescue, which cannot address the initial question. A correct experiment is Elk3 ChIP-seq and Ets3 ChIP-seq. Since the authors also appealed that they cannot perform ChIP-seq on these two factors due to the limitation of cancer tissues, they should change this sentence.*

Indeed, the availability of CSC, which can be isolated from tumors is limiting. However, to answer the question raised by this reviewer, we performed ETS2-CHIP-qPCR in phosphomimetic ETS SCC-SCs in vivo to test whether ETS2 indeed binds to epicenters within SCC-SE. We now show that endogenous ETS2 as well as myc-tagged ETS2 binds to epicenters of SCC-SE including *Cxcl1/2, Ets2* and Il1b (Figure 6). Please also see Figure 6 and our response to Reviewer #1 main point. We’ve also carried out RNA-seq and H3K27ac ChIP-seq on purified basal progenitors from the phosphomimetic induced skin epidermis. The results are beautiful and striking.

*3) The subtitle "Testing the physiological significance of super-enhancer dynamics in SCCs" was over-claimed, please change it accordingly to the data.*

We changed this subheading to “Testing the physiological significance of ETS2 in super-enhancer dynamics” as we now have provided functional data showing that ETS2 is required for maintenance and in sufficient to induce CSC-SE in otherwise normal skin epithelium.

*4) In paragraph two of the subsection “SE-associated ETS TFs constitute cancer vulnerabilities within SCCs”, the authors claimed* Elk3 transcription was knocked down by >90%, while in Figure 3 the efficiency is not that high.

We apologize and have changed it to >70% in the text, although we also now add very nice Western blots to show efficiency.

*5) Similarly, in the fourth paragraph of the aforementioned section, the authors claimed* Ets2 transcription was knocked down by 80-90%, while, in Figure 4, the efficiency does not appear to be that high either.

We apologize and have changed it to >65% in the text, although we add nice Western Blots to show the efficiency.

*6) In "SE-associated ETS TFs constitute cancer vulnerabilities within SCC", the authors started from the purpose to explore whether* Elk3 *and* Ets2 *participate in governing super-enhancer dynamics, but turned out to test the functional significance of them. I would suggest changing the first two sentences to focus more on their function.*

We completely agree and have removed the second sentence: “Given that both of these TFs are known to bind to ETS sequence motifs, we wondered whether they might both participate in governing super-enhancer dynamics. To test this possibility,” and changed it to: “To test their functional relevance, we began by focusing on ELK3…”.

*7) The title of the paper concludes that ETS family transcriptional regulators drive chromatin dynamics. However, in the paper, due to the difficulties to profile global chromatin states by ChIP-seq, no direct proof can be found to fully support this conclusion. Without direct evidence to support the major conclusion of the paper it is not of sufficient impact for eLife. It should be noted that the existence of a binding motif does not necessarily indicate the binding of a transcription factor. The regulation of the recruitment of transcription factors has been shown by many studies.*

We have now added genetic proof that ETS2 indeed regulates global chromatin dynamics in CSC and would like to refer the reviewer to the new Figure 6 and the new paragraphs (please see “Testing the physiological significance of ETS2 in super-enhancer dynamics”).

To further validate our ETS2 overexpression model, we now also added data showing that expression of phosphomimetic constitutive-active ETS2 but not wild-type ETS2 induces hyperproliferation and marked epidermal thickening and invaginations. In addition we also added data showing that constitutive-active ETS2 also increased CD11b+ inflammatory cell recruitment (Figure 5—figure supplement 1) – another hall mark of SCC development.